# Optimization Proxies using Limited Labeled Data and Training Time – A Semi-Supervised Bayesian Neural Network Approach

**Parikshit Pareek** [1]   **Abhijith Jayakumar** [2]   **Kaarthik Sundar** [2]   **Sidhant Misra** [2]   **Deepjyoti Deka** [3]

## Abstract

Constrained optimization problems arise in various engineering systems such as inventory management and power grids. Standard deep neural network (DNN) based machine learning proxies are ineffective in practical settings where labeled data is scarce and training times are limited. We propose a semi-supervised Bayesian Neural Networks (BNNs) based optimization proxy for this complex regime, wherein training commences in a sandwiched fashion, alternating between a supervised learning step for minimizing cost, and an unsupervised learning step for enforcing constraint feasibility. We show that the proposed semi-supervised BNN outperforms DNN architectures on important non-convex constrained optimization problems from energy network operations, achieving up to a tenfold reduction in expected maximum equality gap and halving the inequality gaps. Further, the BNN's ability to provide posterior samples is leveraged to construct practically meaningful probabilistic confidence bounds on performance using a limited validation data, unlike prior methods.

## 1. Introduction

Constrained optimization problems are fundamental in the optimal operation of various engineering systems, such as supply chains, transportation networks, and power grids. Learning a forward mapping between the inputs and outputs of these problems can significantly reduce computational burdens, especially when rapid solutions are required, such as in electricity markets or real-time transportation planning.

Recent advancements in machine learning (ML) have led to considerable efforts to solve optimization problems using deep neural networks (DNNs) (Khadivi et al., 2025; Kotary et al., 2021; Fajemisin et al., 2024). The idea of learning input-to-output mappings has been explored via supervised and unsupervised methods, particularly in power system applications (Zamzam & Baker, 2020; Donti et al., 2021; Park & Van Hentenryck, 2023; Fioretto et al., 2020; Kotary et al., 2021; Rolnick et al., 2022a; Piloto et al., 2024). Additionally, constraint penalization approaches have been proposed to enforce feasibility in predicted outputs within DNN loss functions (AI4OPT, 2023).

Supervised DNN models rely on labeled datasets obtained by solving numerous ($10^4$ or more) instances of optimization problems. This data generation step poses a significant limitation due to prohibitive computational times required for moderate to large problem instances, particularly if the system topology and other parameters change over time. For example, (Park & Van Hentenryck, 2023) report that generating labeled data for a medium-sized power grid problem takes over three hours[1]. Unsupervised methods aim to address the labeled data generation issue (Donti et al., 2021; Park & Van Hentenryck, 2023); however, they often have high training time requirements, particularly due to the use of projection/correction steps to ensure feasibility within the framework (Donti et al., 2021; Gupta et al., 2022; Zamzam & Baker, 2020). Moreover, unsupervised methods still require large number of labeled data to perform validation and provide confidence bounds on error with respect to true solution.

Thus, it is important to note that practical performance of ML based optimization proxies needs to be qualified under constraints of both : (a) `Total Labeled Data` `(training + validation)`, and (b) `Training time`. This is true in the context of bi-level optimization problems such as in power grid planning where ML proxies may serve as subroutines to simulate decision-making processes for given first-level decisions (Ibrahim et al., 2020). Here, ML models must be adaptable in both training and validation to changing problem inputs and parameters. Minimizing or limiting both the time and data required for learning input/output mappings in optimization problems is thus cru-

---

[1]Department of Electrical Engineering, Indian Institute of Technology Roorkee, India [2]Los Alamos National Laboratory, NM, USA [3]MIT Energy Initiative, MIT, Cambridge, USA. Correspondence to: Parikshit Pareek <pareek@ee.iitr.ac.in>.

*Proceedings of the $42^{nd}$ International Conference on Machine Learning*, Vancouver, Canada. PMLR 267, 2025. Copyright 2025 by the author(s).

---

[1]See Table 4 in (Park & Van Hentenryck, 2023).

cial.

Estimating the generalization error of ML models over the testing dataset is another aspect that is particularly relevant to engineered systems where the system must obey physical and safety limits. When limited labeled data is available for validation, one relies on concentration results such as Hoeffding's inequality (Sridharan, 2002; Hoeffding, 1994) to develop error bounds using finite out-of-sample data. However, these bounds are often loose and impractical, and creates the need for frameworks that enable tighter expected error bounds with limited labeled validation data.

**Contributions:** Motivated by the preceding discussion, this paper considers the problem of designing optimization proxies with improved confidence bounds in the setting of limited *training time* requirement and limited labeled data availability. Our major contribution is the development of a Bayesian Neural Network (BNN) coupled with a semi-supervised training approach for this setting, that can be used to give tight confidence bounds on predictions. First, we propose the use of BNNs instead of DNNs for learning input-to-output mappings, as they provide intrinsic *uncertainty quantification* and allow the integration of prior beliefs (Papamarkou et al., 2024). Second, we introduce a Sandwich learning method for BNN, which integrates unlabeled data into training through feasibility-based data augmentation. This approach enforces feasibility without requiring more labeled instances. Third, we utilize the predictive variance information provided by BNNs to develop tight and practically useful expected error bounds using Bernstein concentration bounds (Audibert et al., 2007). We intentionally restrict ourselves to 1000 training instances and 10 minutes of training tim on a single CPU core to demonstrate the effectiveness of the proposed learning scheme under low-data, low-compute settings. For various power grid optimization problem instances (57-Bus, 118-Bus, 500-Bus and 2000-Bus), we show that (i) supervised BNNs outperform standard supervised DNN approaches under limited training time and data; (ii) the proposed Sandwich BNN enforces feasibility better than supervised BNNs without requiring additional training time or data; and (iii) the Bernstein bound-based expected error bounds are tight and practically useful for constraint satisfaction studies without extra computational effort. The proposed BNN-based approaches achieve at least an order of magnitude lower maximum equality gap compared to state-of-the-art DNN models, without compromising the optimality gap.

### 1.1. Related Work

In recent years, Deep Neural Networks (DNNs) have been applied to solve various optimization problems with physics-based constraints, particularly in energy networks (Zamzam & Baker, 2020; Gupta et al., 2022; Donti et al., 2021; Singh

et al., 2021; Park & Van Hentenryck, 2023; Kotary et al., 2021). The primary motivation is to replace time-consuming optimization algorithms with machine learning proxies, enabling instantaneous solutions to a large number of problem instances (Park & Van Hentenryck, 2023; Donti et al., 2021; Gupta et al., 2022; Zamzam & Baker, 2020).

Outside the realm of optimization proxies, several semi-supervised learning methods have been proposed to leverage unlabeled data for improving ML model performance (Yang et al., 2022). These approaches include augmenting unlabeled data with inexpensive pseudo-labels and developing unsupervised loss functions to be minimized alongside supervised loss functions (Sharma et al., 2024; Yang et al., 2022). Data augmentation has been used in image classification with Bayesian Neural Networks (BNNs) using the notion of semantic similarity (Sharma et al., 2024). However, this concept is not readily extensible to ML proxies for constrained optimization problems, where semantic similarity is hard to quantify for input variations leading to changes in the output. To address this challenge, we propose a feasibility-based data augmentation scheme that relates directly to the constraints of the optimization problem. To the best of our knowledge, these ideas have not been explored in the context of BNN algorithms for solving large-scale optimization problems. A related but distinct line of work involves loss function-based prior design for output constraint satisfaction (Sam et al., 2024; Yang et al., 2020).

## 2. Background/Preliminaries

### 2.1. Problem Setup and Assumptions

We consider nonlinear and non-convex, constrained optimization problems involving both equality constraints $g(\mathbf{x}, \mathbf{y}) = 0$ and inequality constraints $h(\mathbf{x}, \mathbf{y}) \leqslant 0$, where $\mathbf{y}$ represents the decision variables and $\mathbf{x}$ represents the input variables, both as vectors. The objective is to minimize the cost function $c(\mathbf{y})$. Mathematically, the optimization problem is as follows:

$$\min_{\mathbf{y}} \quad c(\mathbf{y}) \tag{1a}$$

$$\text{s.t.} \quad g(\mathbf{x}, \mathbf{y}) = 0 \tag{1b}$$

$$h(\mathbf{x}, \mathbf{y}) \leqslant 0 \tag{1c}$$

$$\mathbf{x} \text{ is given (input vector)} \tag{1d}$$

We assume that for all $\forall \mathbf{x} \in \mathcal{X}$, i.e., for any input vector in the set $\mathcal{X}$ ($\mathcal{X}$ could be as simple as a hyper-rectangle), there exists at least one feasible solution to problem (1). Let $\mathcal{D} = \{(\mathbf{x}_i, \mathbf{y}_i^\star)\}_{i=1}^{N}$ denote the labeled dataset, where $\mathbf{y}_i^\star$ is the optimal solution obtained by solving optimization problem (1) for each $\mathbf{x}_i$. Assuming that sampling the input vector $\mathbf{x}$ is inexpensive, we construct an unlabeled dataset $\mathcal{D}^u = \{\mathbf{x}_j\}_{j=1}^{M}$.

Our goal is to develop a BNN surrogate that provides an approximate optimal value of the decision variables $\widehat{\mathbf{y}}_t$ for a given test input vector $\mathbf{x}_t \in \mathcal{X}$. This work falls under the category of developing *optimization proxies* or *surrogates*, where the machine learning model serves as a direct forward mapping between the input and output variables of an optimization problem (see (Park & Van Hentenryck, 2023)).

The paper proposes a semi-supervised framework to solve this problem, wherein training alternates between a supervised step—using labeled data $\mathcal{D}$ to minimize prediction error—and an unsupervised step—using unlabeled data $\mathcal{D}^u$ to enforce the feasibility of constraints in (1b) and (1c). Both steps are implemented using a Bayesian Neural Network (BNN).

### 2.2. Bayesian Neural Network

We consider a Bayesian Neural Network (BNN) denoted as $f_w(\mathbf{x})$, where $w$ represents all the weights and biases of the network. These weights are assigned an isotropic normal prior $p(w)$ with covariance $\sigma^2 I$, meaning that each weight is independently normally distributed with zero mean and variance $\sigma^2$.

In the supervised training of the BNN, the goal is to compute the posterior distribution over the weights given the labeled data $\mathcal{D} \equiv (\mathbf{x}, \mathbf{y})$. This posterior is expressed as $p(w \mid \mathbf{x}, \mathbf{y}) \propto p(\mathbf{y} \mid \mathbf{x}, w)\, p(w)$. Here, $p(\mathbf{y} \mid \mathbf{x}, w)$ is the likelihood of the labeled data given the weights, and $p(w)$ is the prior over the weights. The posterior distribution $p(w \mid \mathbf{x}, \mathbf{y})$ encapsulates the uncertainty about the weights after observing the data. Due to computational challenges in calculating the normalization constant of the posterior, approximate methods such as stochastic variational inference (SVI) with the mean-field assumption are employed the posterior distribution estimation (see (Jospin et al., 2022)).

For making predictions, the posterior predictive distribution is approximated as $p(\mathbf{y}^t \mid \mathbf{x}^t, \mathcal{D}) = \mathbb{E}_{p(w|\mathcal{D})}\left[p\big(f_w(\mathbf{x}^t)\big)\right]$, where $\mathbf{x}^t$ is a test input vector, and the expectation is taken over the approximate posterior distribution of the weights. Moreover, we assume a Gaussian likelihood for output:

$$p(\mathbf{y} \mid \mathbf{x}, w) = \prod_i \mathcal{N}\left(\mathbf{y}_i \mid f_w(\mathbf{x}_i), \sigma_s^2\right),$$

with $\sigma_s^2$ being a parameter in the SVI that controls the spread (noise variance) around the target values, and $(\mathbf{x}_i, \mathbf{y}_i) \in \mathcal{D}$. Adapting this approach to update the BNN using the unsupervised data $\mathcal{D}^u$ requires the definition of a suitable likelihood function, detailed in the next section, along with the semi-supervised framework to obtain the BNN surrogate.

## 3. Semi-supervised Learning: Sandwich BNN

We start by defining a suitable likelihood function for the unsupervised learning process. To that end, we augment the unlabeled data $\mathcal{D}^u$ using the necessary feasibility conditions that the vector $\mathbf{y}$ must satisfy to be a solution of (1). We propose a function $\mathcal{F}(\mathbf{y}, \mathbf{x})$ to measure the feasibility of a candidate solution $\mathbf{y}$ for a given input $\mathbf{x}$. This function consists of two terms: one measuring the equality gap and the other measuring the one-sided inequality gap or violations. The relative emphasis on each term is determined by the parameters $\lambda_e$ and $\lambda_i$, respectively, i.e.,

$$\mathcal{F}(\mathbf{y}, \mathbf{x}) = \lambda_e \underbrace{\left\|g(\mathbf{x}, \mathbf{y})\right\|^2}_{\texttt{Equality Gap}} + \lambda_i \underbrace{\left\|\text{ReLU}[h(\mathbf{x}, \mathbf{y})]\right\|^2}_{\texttt{Inequality Gap}}. \quad (2)$$

For any given feasible solution $\mathbf{y}_c$ for the optimization problem in (1)[2], we have $\mathcal{F}(\mathbf{y}_c, \mathbf{x}) = 0$ for the given input $\mathbf{x} \in \mathcal{X}$. Furthermore, because of our assumption in Sec. 2.1 that the problem in (1) has at least one feasible solution, the minimum value $\mathcal{F}(\cdot, \mathbf{x}) = 0$ for any $\mathbf{x} \in \mathcal{X}$. Therefore, we can augment the unlabeled dataset $\mathcal{D}^u$ to create a labeled feasibility dataset, i.e., $\mathcal{D}^f = \{(\mathbf{x}_j, \mathcal{F}(\cdot, \mathbf{x}) = 0)\}_{j=1}^M$. Since input sampling is inexpensive, constructing this feasibility dataset $\mathcal{D}^f$ incurs no additional computational cost. Similar to the supervised step in Sec. 2.2, we now define a Gaussian likelihood for the unsupervised training step, with $\sigma_u^2$ as the noise variance for unsupervised learning and $\mathbf{x}_j \in \mathcal{D}^f$, as

$$p(\mathcal{F} \mid \mathbf{x}, w) = \prod_j \mathcal{N}\left(0 \mid \mathcal{F}\big(f_w(\mathbf{x}_j), \mathbf{x}_j\big), \sigma_u^2\right),$$

To obtain an optimization proxy, we parameterize the candidate solution $f_w(\mathbf{x})$ using a Deep Neural Network (DNN)-style architecture and employ a sandwich-style semi-supervised training for the BNN, as illustrated in Figure 1. The fundamental idea of this training method is to update the network weights and biases through multiple rounds of training in which each round alternatives between using the labeled dataset $\mathcal{D}$ for prediction or cost optimality, and the augmented feasibility data set $\mathcal{D}^f$ for constraint feasibility. We let *Sup* and *UnSup* denote the inference steps in the BNN training using $\mathcal{D}$ and $\mathcal{D}^f$, respectively. Both *Sup* and *UnSup* are performed for a fixed maximum time, with the total training time constrained to $T_{\max}$. Finally, the prediction of the mean estimate $\mathbb{E}_{\mathbf{y}^t}$ and the predictive variance estimate $\mathbb{V}_{\mathbf{y}^t}$ is accomplished using an unbiased Monte Carlo estimator by sampling 500 weights from the final weight posterior $p_W^m$.

### 3.1. Selection via Posterior (SvP)

In Bayesian Neural Network (BNN) literature, the standard approach is to use the mean posterior prediction

---

[2]Not necessarily optimal for (1).

*Figure 1.* Flowchart of the proposed sandwich-style BNN learning. The *Sup* block represents the supervised learning stage with labeled dataset $\mathcal{D}$, and the *UnSup* block represents the unsupervised learning with the augmented feasibility dataset $\mathcal{D}^f$. Learning time upper limits are represented as $T_s$, $T_u$, and $T_{\max}$ for *Sup*, *UnSup*, and the complete semi-supervised BNN learning, respectively. At the prediction stage, $\mathbf{Y}$ denotes the posterior prediction matrix (PPM) for one test input sample, where each column represents the predicted output obtained via one weight sample from the posterior.

$\mathbb{E}_{p(w|\mathcal{D})}[f_w(\mathbf{x}^t)]$ for a test input $\mathbf{x}^t$. This is similar to using the mean prediction of ensemble Deep Neural Networks (DNNs). However, unlike DNNs, BNNs can provide multiple predictions without additional training cost, as we can sample multiple weight instances from the posterior distribution $p_W^m$ and construct the posterior prediction matrix (PPM) $\mathbf{Y}$ (see Figure 1 for details and Appendix B.3 for structure of the PPM.). We propose to use the PPM to improve the feasibility of the predicted output of the optimization proxy. Each column of the PPM represents one predicted output vector corresponding to a weight sample. We select the weight sample $W^\star$ that minimizes the maximum equality gap, defined as:

$$W^\star = \arg\min_j \left[ \max_i \left| g_i(\mathbf{x}^t, \mathbf{Y}_{\cdot j}) \right| \right], \qquad (3)$$

where $\mathbf{Y}_{\cdot j}$ is the $j$-th column of the PPM, and $g_i(\cdot, \cdot)$ represents the $i$-th equality constraint function.[3] The output prediction corresponding to the weight sample $W^\star$ will have the minimum equality gap, and we term this process *Selection via Posterior* (SvP). Note that the numerical operation in (3) can be performed in parallel and has minimal computational cost compared to analytical projection methods in (Zamzam & Baker, 2020) that focus on projecting the prediction onto (1c) to satisfy the inequality constraints in the problem (1). Note that it is an application motivated design choice to emphasize enforcement of the equality gap by using the SvP in (3). This can easily be adapted to account for inequality constraints without any significant computational overhead.

## 4. Probabilistic Confidence Bounds

This section focuses on providing bounds on the expected absolute error of our method, i.e., testing error. We explore probabilistic confidence bounds (PCBs) for optimization proxies. The core concept of PCBs is to first evaluate models on a labeled testing dataset with $M$ samples, compute the empirical mean error, and then probabilistically bound

the error for any new input. Specifically, PCBs assert that the expected error will be within $\varepsilon$ of the empirical errors computed from $M$ **out-of-sample** inputs, with a high probability (usually 0.95). Mathematically, we aim to provide a guarantee on the error $e = y - y^t$, where $y^t$ is the BNN prediction and $y$ is the true value, as

$$\mathbb{P}\left\{ \left| \mathbb{E}[|e|] - \frac{1}{M} \sum_{i=1}^{M} |e_i| \right| \leqslant \varepsilon \right\} \geqslant 1 - \delta \qquad (4)$$

where $\mathbb{E}[|e|]$ represents the expected absolute error, $1 - \delta$ is the confidence level, and $\varepsilon$ is the allowable prediction error.

Ideally, we would like to evaluate our model on a large number of samples since, as $M \to \infty$, the error bound $\varepsilon \to 0$. However, increasing $M$ leads to a higher requirement for labeled data, which defeats the purpose of training using low labeled data [4]. To address this issue, confidence inequalities are commonly used to provide PCBs, with Hoeffding's inequality ((Hoeffding, 1994)) being one of the most widely used bounds. As stated in Appendix E, Hoeffding's inequality assumes that the error is bounded (i.e., $|e_i| \leqslant R$ for all $i$) and provides PCBs whose tightness is governed by $M$, with the relationship $\varepsilon = R\sqrt{\frac{\log(2/\delta)}{2M}}$. However, the Hoeffding's bound can often be too loose to be practically relevant.

To improve upon this, we propose using Bernstein's inequality (see (Audibert et al., 2007)) as the concentration bound, which utilizes the *total variance in error* (TVE) information along with $M$, under the same bounded error assumption. The main challenge in using the Bernstein bound is obtaining the TVE without extensive out-of-sample testing. One possible solution is to use the empirical Bernstein bound as in (Audibert et al., 2007; Mnih et al., 2008), which employs the *empirical variance of the error* $\widehat{\mathbb{V}}_e$, obtained from the same $M$ testing samples, and accounts for the error in variance estimation by modifying the theoretical Bernstein inequality. Mathematically, the PCB using the Empirical

---

[3] In a general setting of constrained optimization problems, there may be multiple equality and inequality constraints.

[4] Note that the total labeled data requirement is the sum of training and testing samples, i.e., $N + M$

Bernstein inequality is

$$\varepsilon = \sqrt{\frac{2\widehat{\mathbb{V}}_e \log{(3/\delta)}}{M}} + \frac{3R\log{(3/\delta)}}{M},$$

as given in (Audibert et al., 2007; Mnih et al., 2008). Since the term under the square root depends on the empirical TVE $\widehat{\mathbb{V}}_e$ rather than $R$, the empirical Bernstein bound becomes tighter more quickly with increasing $M$ if $\widehat{\mathbb{V}}_e \ll R$.

To further tighen the bound, we propose using the Theoretical Bernstein bound (Sridharan, 2002) (Theorem E.3 in Appendix E) with the *Mean Predictive Variance* (MPV) as a proxy for the TVE $\mathbb{V}_e$. The MPV is the mean of the predictive variance of testing samples, i.e., MPV $= \frac{1}{M}\sum_{k=1}^{M} \mathbb{V}_W\left[\mathbf{Y}_{i:}^k\right]$, where variance $\mathbb{V}_W\left[\mathbf{Y}_{i:}^k\right]$ for the $k^{th}$ test sample using entries of the posterior prediction matrix across columns, generated using posterior weights, for $i^{th}$ output variable. In principle, the MPV captures the expected variance in the predictions due to the posterior distribution of BNN weights. We hypothesize that with a constant multiplier $\alpha > 1$,

$$\alpha\, \mathrm{MPV} \geqslant \mathbb{V}_e = \mathbb{E}_M\left[\mathbb{V}_W[e]\right] + \mathbb{V}_W\left[\mathbb{E}_M[e]\right] \geqslant \mathbb{V}_{|e|}, \quad (5)$$

where $\mathbb{E}_M$ and $\mathbb{E}_W$ denote expectations with respect to $M$ testing samples and posterior weight samples, respectively, and $\mathbb{V}_M[e]$ and $\mathbb{V}_W[e]$ denote the variance of the error with respect to $M$ testing samples and posterior weight samples, respectively. The equality in (5) follows from the law of total variance (Blitzstein & Hwang, 2019). $\mathbb{V}_{|e|}$ represents the variance of the absolute value of the error, which is lower than the variance of the error.

Notice that the first term of the TVE, $\mathbb{E}_M\left[\mathbb{V}_W[e]\right]$, is independent of the labeled testing dataset because the true output $y$ is constant with respect to posterior weight samples; thus, $\mathbb{V}_W[e] = \mathbb{V}_W[y - y^t] = \mathbb{V}_W[y^t]$. Furthermore, from the definition of MPV, we have MPV $= \mathbb{E}_M\left[\mathbb{V}_W[e]\right]$. As an example, if $\mathbb{V}_W\left[\mathbb{E}_M[e]\right] \leqslant \mathrm{MPV}$, our hypothesis in (5) holds with $\alpha = 2$. Consequently, we can use $2 \times \mathrm{MPV}$ as an upper bound for $\mathbb{V}_e$ in the Theoretical Bernstein bound (see Theorem E.3 in Appendix E), which gives the error bound

$$\varepsilon = \sqrt{\frac{4 \times \mathrm{MPV}\log{(1/\delta)}}{M}} + \frac{2R\log{(1/\delta)}}{3M},$$

which is better than the Empirical Bernstein bounds. The hypothesis in (5) and the corresponding constant $\alpha$ can be computed by using application specific information or performing a meta-study like in Section 5.

The strength of this approach is that we do not require labeled testing samples to calculate MPV, thus incurring no additional computational burden from generating labels. Also, note that this constitutes an advantage of BNNs since MPV information is readily available with BNNs but cannot be obtained from DNN-based optimization proxies.

In the next section, we perform a meta-study using different BNN models on different test cases to show the performance of our proposed learning architecture as well as demonstrate that hypothesis (5) indeed holds for the proposed optimization proxy learning problems.

## 5. Numerical Results and Discussion

We test the proposed method on the Alternating Current Optimal Power Flow (ACOPF) problem, essential for the economic operation of electrical power grids (Molzahn et al., 2019). Efficient ACOPF proxies can mitigate climate change by enabling higher renewable energy integration, improving system efficiency by minimizing losses and emissions, and enhancing grid resiliency against extreme weather conditions (Rolnick et al., 2022a). ACOPF is a constrained optimization problem with nonlinear equality constraints and double-sided inequality bounds. It aims to find the most cost-effective generator set points while satisfying demand and adhering to physical and engineering constraints. The inputs are active and reactive power demands; outputs include generator settings, voltage magnitudes, and phase angles at each bus. We adopt the standard ACOPF formulation (Babaeinejadsarookolaee et al., 2019; Park & Van Hentenryck, 2023; Coffrin et al., 2018) and benchmark our method using the open-source `OPFDataset` from `Torch_Geometric`, which provides numerous solved ACOPF instances (see (Lovett et al., 2024)).

In our results, 'Gap%' denotes the average relative optimality gap compared to the objective values in the labeled testing instances. 'Max Eq.' and 'Mean Eq.' represent the maximum and mean equality gaps over all equality constraints, while 'Max Ineq.' and 'Mean Ineq.' indicate the same for inequality gaps in per unit, all averaged over testing instances. We compare our proposed method with the following state-of-the-art baseline supervised learning models available in the literature using the same labeled dataset and training time constraints, utilizing AI4OPT's ML4OPF package (AI4OPT, 2023) (network and hyper-parameter details are in Appendix B):

- **Naïve MAE and Naïve MSE (Supervised)**: Use $l_1$-norm and $l_2$-norm loss functions, respectively, to measure differences between predicted and actual optimal solutions (Park & Van Hentenryck, 2023), incorporating a *bound repair layer* with a `sigmoid` function. The bound repair layer ensures that inequality constraints are always satisfied.

- **MAE + Penalty, MSE + Penalty, and LD (Supervised)**: Add penalty terms for constraint violations

*Table 1.* Comparative performance results for the ACOPF Problem for 'case57' with 512 labeled training samples, 2048 unlabeled samples, and $T_{max} = 600$ sec.

| Method | Gap% | Max Eq. | Mean Eq. | Max Ineq. | Mean Ineq. |
|---|---|---|---|---|---|
| Sandwich BNN SvP (Ours) | **0.928** | **0.027** | 0.006 | **0.000** | **0.000** |
| Sandwich BNN (Ours) | 0.964 | 0.045 | **0.005** | 0.000 | 0.000 |
| Supervised BNN SvP (Ours) | 3.195 | 0.083 | 0.011 | 0.000 | 0.000 |
| Supervised BNN (Ours) | 3.255 | 0.130 | 0.011 | 0.000 | 0.000 |
| Sandwich DNN (Ours) | 2.878 | 0.358 | 0.014 | 0.006 | 0.000 |
| Naïve MAE | 4.029 | 0.518 | 0.057 | 0.000 | 0.000 |
| Naïve MSE | 3.297 | 0.541 | 0.075 | 0.000 | 0.000 |
| MAE + Penalty | 3.918 | 0.370 | 0.037 | 0.000 | 0.000 |
| MSE + Penalty | 3.748 | 0.298 | 0.039 | 0.000 | 0.000 |
| LD + MAE | 3.709 | 0.221 | 0.033 | 0.000 | 0.000 |

to the naïve MAE or MSE loss functions (Park & Van Hentenryck, 2023). The Lagrangian Duality (LD) method applies the $l_1$-norm as outlined in (Fioretto et al., 2020; Park & Van Hentenryck, 2023), and also uses a *bound repair layer* with a `sigmoid` function.

We exclude self-supervised constrained optimization methods like Primal-Dual Learning (PDL) (Park & Van Hentenryck, 2023) and DC3 (Donti et al., 2021) due to their significantly higher training times and computational demands, which violate the premise of this paper. For example, PDL requires over 125 minutes of training time for the ACOPF problem on a 118-node power network using a Tesla RTX6000 GPU. Methods that require solving alternating current power flow to recover solutions (e.g., (Zamzam & Baker, 2020)) are also excluded, as they result in high prediction times compared to DNN or BNN forward passes[5]. Furthermore, Graph Neural Network-based large models, such as (Piloto et al., 2024), require extensive training datasets—for instance, (Piloto et al., 2024) utilizes 270k training samples.

To demonstrate the effectiveness of the proposed BNN learning methods, we conduct simulation studies on both our models and the `ML4OPF` models using an M1 Max CPU with 32 GB RAM, without any GPU. This setup highlights performance improvements due to the learning mechanism rather than computational power. We propose two classes of different models as:

- **Supervised BNN and Supervised BNN SvP**: Standard BNN learning with labeled data, utilizing mean prediction and Selection via Posterior (SvP), respectively. The network uses `ReLU` activation and **no** *bound repair layer*.

- **Sandwich BNN and Sandwich BNN SvP**: The proposed Sandwich BNN trained with labeled and unlabeled data as discussed in Section 3. Unsupervised training utilizes four times the number of labeled data samples. The network uses `ReLU` activation and **no** *bound repair layer*.

Importantly, we intentionally keep the BNN architecture unoptimized, constructing it using four different sub-networks (one for each of the four ACOPF outputs: real power generation, reactive power generation, voltage magnitude, and voltage angle). Each sub-network has `two hidden layers` with the number of hidden neurons equal to `2 × input size`. In the *Sup* stage of the Sandwich BNN, both weights and biases are updated, whereas in the *UnSup* stage, only the weights are modified via SVI. Best model out of five random trials is selected.

Tables 1 and 2 present the comparative performance of different methods for solving the ACOPF problem on the 'case57' and 'case118' test cases, containing 57 and 118 nodes, respectively. For 'case57', the Sandwich BNN SvP method achieves the best Max Eq. performance (**0.027**), outperforming all other methods, including the standard Sandwich BNN, without compromising other metrics. All the methods proposed in this paper outperform the best DNN results, typically achieved with the LD+MAE model (last row in the tables). Similar trends are observed for the 'case118' as well.[6] It is important to contextualize the significance of these numerical improvements. In ACOPF problems, cost values are in USD, with the mean cost for 'case118' being $97,000 or 9.7 in the per-unit system. Therefore, a 1% Gap corresponds to an expected difference of $970 across the testing instances. A 'Max Eq.' value of 0.08 implies a maximum expected power imbalance of 8.0 Megawatts among all 118 nodes of 'case118'. Thus, reducing the Max Eq. from 1.284 with the LD+MAE model to 0.089 with our Sandwich BNN SvP model represents a significant improvement.

---

[5]see prediction time studies in (Donti et al., 2021) which suggest 10 times higher prediction time with power flow based projections (0̃.080 sec. compared to 0.001 sec. for DNN and 0.003 sec. per testing instance for proposed BNNs).

[6]See Appendix C for similar tabular results and discussion of trends on larger test instances.

*Table 2.* Comparative performance results for the ACOPF Problem for 'case118' with 512 labeled training samples, 2048 unlabeled samples, and $T_{max} = 600$ sec.

| Method | Gap% | Max Eq. | Mean Eq. | Max Ineq. | Mean Ineq. |
|---|---|---|---|---|---|
| Sandwich BNN SvP (Ours) | **1.484** | **0.089** | 0.018 | 0.008 | **0.000** |
| Sandwich BNN (Ours) | 1.485 | 0.100 | **0.016** | 0.008 | 0.000 |
| Supervised BNN SvP (Ours) | 1.568 | 0.147 | 0.022 | 0.013 | 0.000 |
| Supervised BNN (Ours) | 1.567 | 0.205 | 0.020 | 0.013 | 0.000 |
| Sandwich DNN (Ours) | 3.298 | 0.585 | 0.034 | 0.158 | 0.000 |
| Naïve MAE | 1.638 | 2.166 | 0.187 | **0.000** | 0.000 |
| Naïve MSE | 1.622 | 3.780 | 0.242 | 0.000 | 0.000 |
| MAE + Penalty | 1.577 | 1.463 | 0.102 | 0.000 | 0.000 |
| MSE + Penalty | 1.563 | 2.637 | 0.125 | 0.000 | 0.000 |
| LD + MAE | 1.565 | 1.284 | 0.083 | 0.000 | 0.000 |

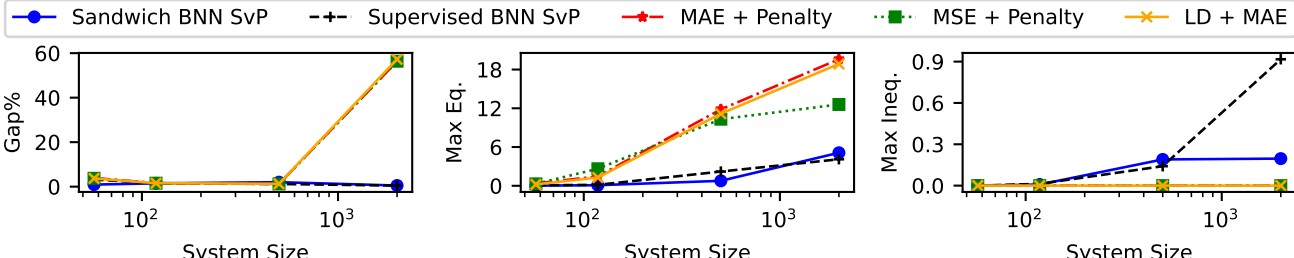

*Figure 2.* Growth trajectories of performance metrics for ACOPF across system sizes for different methods. Detailed results for 'case500' and 'case2000' are given with Table 8 and Table 9 respectively, in Appendix C.

Sandwich DNN in these tables is a DNN with the same network architecture as the BNN, trained under the same time constraints. This network is trained by alternating between minimizing a supervised MSE loss and and an unsupervised feasibility loss as in (2). In both these rounds, an $\ell_1$ weight regularization is added to prevent over-fitting. We see that this DNN model shows performance comparable to other DNN approaches. This is a strong indication that the superior performance of the sandwich BNN model is largely due to the superior performance of the BNN approach in the low data regime.

Figure 2 illustrates the growth of various metrics with increasing system size while keeping training resources constant. The proposed BNN methods exhibit significantly lower scaling in the expected maximum equality gap. Although Sandwich BNN SvP shows slightly higher scaling inequality gaps, the relative improvement in Max Eq. far outweighs these minor drawbacks. Notably, for 'case500' (see Table 8 in the Appendix C), the expected maximum power imbalance is below 0.5% of the mean real power demand ($1.7 \times 10^4$ MW) and power grids already are equipped with spinning reserves (see (Ela et al., 2011)) that have reserve capacity to handle these imbalances. This is a significant improvement over the DNN models which have much higher 'Mean Eq.' values. Moreover, the 'Max Ineq.' growth could be easily suppressed by incorporating *bound repair* layers, as used in DNN models in ML4OPF (AI4OPT, 2023).

We perform robustness study on 'case118', by running five different trials of 10-min and 15-min training time with 512,1024, and 5048 supervised training samples keeping 2048 unsupervised samples constant across models and trials (See Appendix C for detailed graphs and tables). The results clearly shows that increasing training time and number of supervised samples decreases decreases %Gap, Max Eq. and Max Ineq., consistently, and keep Mean Eq. one order of magnitude lower and Mean Ineq. at order $10^{-5}$. The Sandwich BNN SvP model, provides 1.50% Gap, 0.094 as Max Eq. and 0.014 as Max Ineq. with 512 supervised training samples and 10-Min training time,, on average over five trials. The same model provides 1.46% Gap, 0.080 Max Eq. and 0.011 as Max Ineq. with 2048 supervised training samples and 15-Min Training time. This implies that five minute increase in training time with four times more training samples worth 2.66% reduction in optimality error (%Gap), 17.5% reduction in equality constraint feasibility error (Max Eq.) and more than 27% reduction in inequality constraint feasibility error (Max Ineq.).

This detailed study also highlight that smaller training times are not good enough to extract complete information from larger training datasets as various models with 2048 supervised samples perform better with 15-min training come compared to 10-min training time. For instance, Max Eq. is 6.18% lesser for 2048 samples when we increase training time from 10-Min to 15-Min, and %Gap values reduces

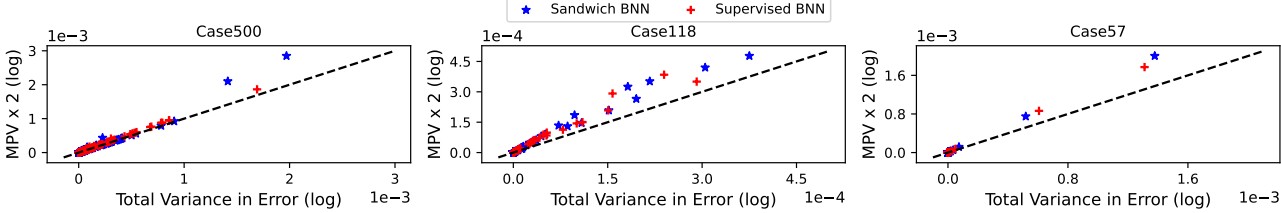

*Figure 3.* Empirical study comparing total variance in error $\widehat{\mathbb{V}}_e$ with $2 \times$ MPV across different cases of ACOPF and the proposed learning mechanisms.

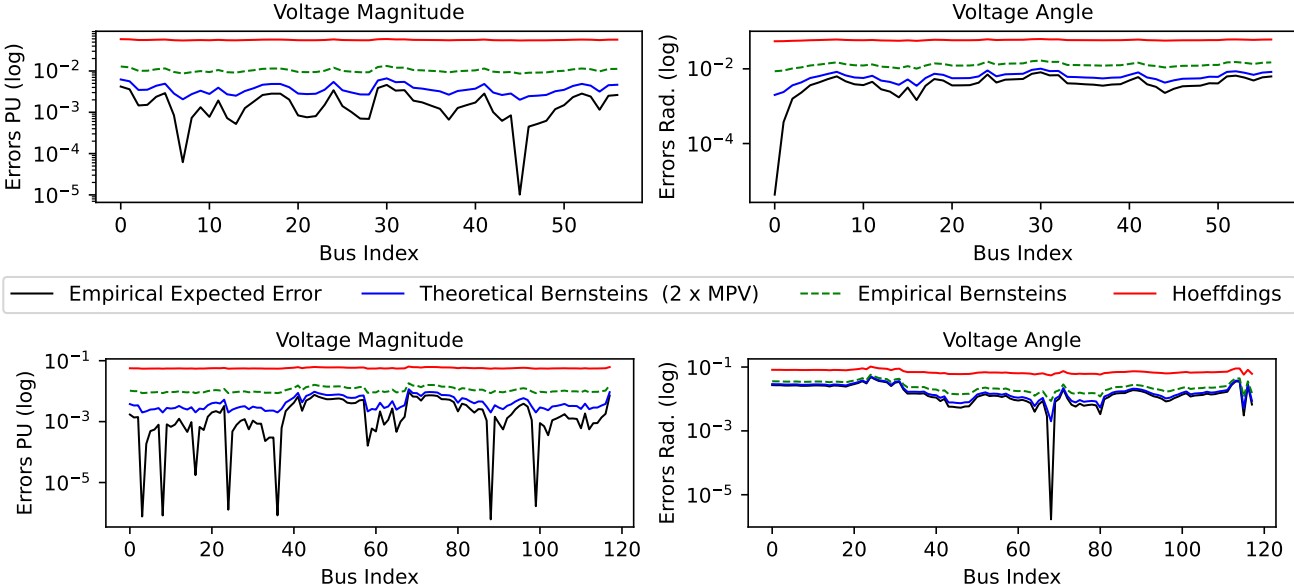

*Figure 4.* Comparison of voltage magnitude and voltage angle error bounds (in logarithmic scale) across bus indices for 'case57' (top row) and 'case118' (bottom row). The plot illustrates that PCBs using theoretical Bernstein bounds with $2 \times$ MPV from hypothesis (5) are tightest among all PCBs. We consider $\delta = 0.95$ and 1000 *out-of-sample* testing data points i.e. $M = 1000$.

from 1.52 to 1.46, on average. Lastly, we also observe that variability in training quality reduces with increasing number of supervised samples and training time.

Next, we present results for Proabilistic Confidence bounds, described in Section 4. Figure 3 shows that $2 \cdot$ MPV consistently serves as an empirical upper bound for total variance in error, validating its robustness across models and system configurations[7]. In Figure 4, the theoretical Bernstein bounds using $\mathbb{V}_e = 2 \cdot$ MPV provide tight, practical bounds, whereas Hoeffding's bounds are overly conservative and not useful for grid operations. For example, in 'case118', the Bernstein bound ensures a probabilistic guarantee on voltage constraint satisfaction, such that a predicted voltage between 0.91–1.09 pu guarantees no violations within the ACOPF limits of 0.90–1.10 pu, i.e., the maximum value of the Bernstein bound on the error is 0.010 pu across all nodes.

Compared to Hoeffding's bound (0.064 pu) or the empirical Bernstein bound (0.018 pu), the Bernstein bound (0.010 pu) is far tighter and more practical, highlighting the benefits of BNNs for optimization proxies. The error bounds for the 'case500' is provided in Fig. 7 of the Appendix C. Finally, we note that both Hoeffding's and empirical Bernstein bounds can also be obtained by testing DNN models across $M$ samples[8].

## 6. Conclusions

In conclusion, this paper introduces a semi-supervised Bayesian Neural Network (BNN) approach to address the challenges of high labeled data requirements and limited training time in learning input-to-output maps for con-

---

[7]Total variance in error is assumed to stabilizing with 1000 testing samples (see Figure 6 in Appendix C).

[8]A form of MPV can be obtained via Ensemble DNNs (Ganaie et al., 2022), however, it will lead to very high computational requirement compared to the BNN.

strained optimization problems. The proposed Sandwich BNN method incorporates unlabeled data through input data augmentation, ensuring constraint feasibility without relying on a large number of labeled instances. We provide tight confidence bounds by utilizing Bernstein's inequality, enhancing the method's practical applicability. Results show that BNNs outperform DNNs in low-data, low-compute settings, and the Sandwich BNN more effectively enforces feasibility without additional computational costs compared to supervised BNNs.

## Reproducibility

We use the open-source ACOPF datasets, provided with `Torch_geometric` (Lovett et al., 2024), to train and test our models as well as standard DNN models. Furthermore, the beginning of Section 5 provides details of the DNN models used to compare the performance of the proposed methods. These models are available in AI4OPT's open-source `ML4OPF` package (AI4OPT, 2023). Hyper-parameter details for these models and the proposed methods are provided in Appendix B. The code used in this paper can be found at `https://github.com/kaarthiksundar/BNN-OPF/`. Additional experimental results can be found in the Supplementary Information.

## Impact Statement

This paper presents work whose goal is to advance the field of Machine Learning. In particular, enhancing the solution of optimization problems will result in more efficient resource utilization, helping industries lower costs and reduce environmental impact. Additionally, refining the ACOPF solution pipeline will play a crucial role in addressing climate change by optimizing renewable energy usage and ensuring the reliable operation of the power grid (Rolnick et al., 2022b).

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

# Supplementary Information

## Optimization Proxies using Limited Labeled Data and Training Time – A Semi-Supervised Bayesian Neural Network Approach

## A. ACOPF Problem: Modeling and Dataset

The alternating current optimal power flow (ACOPF) problem is essential for power grid operations and planning across various time scales. It determines generator set-points for real and reactive power that minimize generation costs while meeting power demand and satisfying physical and operational constraints on output variables. We follow the ACOPF model given in `PowerModels` (Coffrin et al., 2018), and take the dataset from `Torch_geometric` ((Lovett et al., 2024)).

*Table 3.* Sets for ACOPF

| | |
|---|---|
| $N$: buses | $G$: generators, generators at bus $i$ ($G_i$) |
| $E$: branches (forward and reverse orientation, $E_R$) | $S$: shunts, shunts at bus $i$ ($S_i$) |
| $L$: loads, loads at bus $i$ ($L_i$) | $R$: reference buses |

*Table 4.* Data for ACOPF Problem

| **Data** | |
|---|---|
| **Symbol** | **Description** |
| $S_g^l, S_g^u \ \forall k \in G$ | Generator complex power bounds. |
| $c_2^k, c_1^k, c_0^k \ \forall k \in G$ | Generator cost components. |
| $v_l^i, v_u^i \ \forall i \in N$ | Voltage bounds. |
| $S_d^k \ \forall k \in L$ | Load complex power consumption. |
| $Y_s^k \ \forall k \in S$ | Bus shunt admittance. |
| $Y_{ij}, Y_c^{ij}, Y_c^{ji} \ \forall (i,j) \in E$ | Branch $\pi$-section parameters. |
| $T_{ij} \ \forall (i,j) \in E$ | Branch complex transformation ratio. |
| $s_u^{ij} \ \forall (i,j) \in E$ | Branch apparent power limit. |
| $i_u^{ij} \ \forall (i,j) \in E$ | Branch current limit. |
| $\theta_l^{ij}, \theta_u^{ij} \ \forall (i,j) \in E$ | Branch voltage angle difference bounds. |
| **Variables** | |
| **Symbol** | **Description** |
| $S_g^k \ \forall k \in G$ | Generator complex power dispatch. |
| $V_i \ \forall i \in N \ (|V_i| \angle \theta)$ | Bus complex voltage (Magnitude$\angle$Angle). |
| $S_{ij} \ \forall (i,j) \in E \cup E_R$ | Branch complex power flow. |

*Table 5.* Objective Function and Constraints

| **Description** | **Equation** |
|---|---|
| Objective | $\min \left( \sum_{k \in G} \left( c_2^k (S_g^k)^2 + c_1^k S_g^k + c_0^k \right) \right)$ |
| Reference bus voltage angle | $\angle V_r = 0 \quad \forall r \in R$ |
| Generator power bounds | $S_g^l \le S_g^k \le S_g^u \quad \forall k \in G$ |
| Bus voltage bounds | $v_l^i \le |V_i| \le v_u^i \quad \forall i \in N$ |
| Power balance $\forall i \in N$ | $\sum_{k \in G_i} S_g^k - \sum_{k \in L_i} S_d^k - \sum_{k \in S_i} Y_s^k |V_i|^2 = \sum_{(i,j) \in E \cup E_R} S_{ij}$ |
| Branch power flow $\forall (i,j) \in E$ | $S_{ij} = Y_{ij}^* |V_i|^2 + Y_{ij}^* V_i V_j^* / T_{ij} \ ; \ S_{ji} = Y_{ji}^* |V_j|^2 + Y_{ji}^* V_i^* V_j / T_{ij}^*$ |
| Branch apparent power limits | $|S_{ij}| \le s_u^{ij} \quad \forall (i,j) \in E \cup E_R$ |
| Branch current limits | $|S_{ij}| \le |V_i| i_u^{ij} \quad \forall (i,j) \in E \cup E_R$ |
| Branch angle difference bounds | $\theta_l^{ij} \le \angle(V_i V_j^*) \le \theta_u^{ij} \quad \forall (i,j) \in E$ |

# B. Experimental Setting Details

## B.1. Architectures

**Supervised BNN and Sandwich BNN**: For ACOPF problem we use real and reactive power demands as input while predicting all decision and state variables using separate networks

- **Input** real and reactive power demands: Two times the number of nodes having non-zero load

- **Output** real and reactive power generation setpoints, voltage magnitude and voltage angle at each node: Two times the number of generators + Two times the number of nodes

## B.2. Hyper-parameters

### B.2.1. ML4OPF:

`config`: The optimization is performed using the Adam optimizer with a learning rate of $1 \times 10^{-4}$ (taken from (Park & Van Hentenryck, 2023)). All networks have two hidden layers, each with a size of $2 \times$ Number of outputs. For hidden layers, ReLU activation function is selected while, the bound repair mechanism is handled using a `Sigmoid` function (Park & Van Hentenryck, 2023).

`penalty_config`: Multiplier of $1 \times 10^{-2}$, with no excluded keys.

`ldf_config`: Step size $1 \times 10^{-2}$, and a kick-in value of 0. The LDF update frequency is set to 1, and no keys are excluded from the updates.

All other hyperparameters are set at default `ML4OPF` values (AI4OPT, 2023).

### B.2.2. PROPOSED BNNS

For all simulations, maximum training time 10 min., $T_{max} = 600$ seconds, per round time $T_r$ is 200 seconds with 40-60 split between *Sup* and *UnSup* models ($T_s = 80$ amd $T_u = 120$ seconds). For VI, we use `MeanFieldELBO` loss function from `Numpyro` and Adam optimizer with a initial learning rate rate of $1 \times 10^{-3}$, and decay rate of $1 \times 10^{-4}$, both of which are selected via grid search.

We use decay schedule as initial learning rate/$(1 + \text{decay rate} \times \text{step})$ within a *Sup* or *UnSup* model, while simple step decay among different rounds of Sandwich BNN. Learning is done with single batch for all models and each of the BNN weight and bias is parameterized using mean and variance parameters of a Gaussian distribution, with mean field assumption i.e. independent from each other. The prior for each weight and bias has zero mean and $10^{-2}$ variance while likelihood noise variance is initialized with $10^{-5}$ mean and $10^{-6}$ variance for *Sup* and fixed at $10^{-10}$ for *UnSup*.

## B.3. Structure of posterior prediction matrix (PPM)

$$\mathbf{Y} \equiv \begin{bmatrix} y_{11} & \cdots\cdots\cdots & y_{1H} \\ \vdots & \ddots\ddots\ddots & \vdots \\ \cdots & y_{\texttt{Variable,Sample}} & \cdots \\ \vdots & \ddots\ddots\ddots & \vdots \\ y_{O1} & \cdots\cdots\cdots & y_{OH} \end{bmatrix} \qquad \begin{bmatrix} O : \text{Number of variables} \\ H : \text{Number of posterior samples} \end{bmatrix}$$

## C. Robustness Study on 118-System

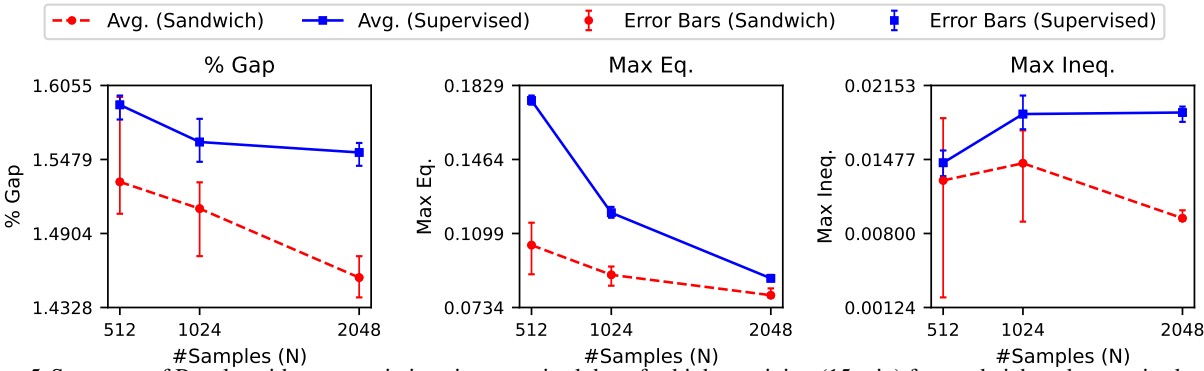

*Figure 5.* Summary of Results with more variations in supervised data, for higher training (15 min) for sandwich and supervised models with SvP. Note that this computation was done on a Max Mini M4 with 24GB RAM.

*Table 6.* Comparison of Average Values (Over Five Trials) for BNN Models: 10/15 min, Sandwich SvP/Supervised SvP. Note that this computation was done on a Max Mini M4 with 24GB RAM.

| N | Sandwich BNN SvP, 15-Min | | | | | Supervised BNN SvP, 15-Min | | | | |
|---|---|---|---|---|---|---|---|---|---|---|
| | %Cost | Max Eq | Mean Eq | Max Ineq | Mean Ineq | %Cost | Max Eq | Mean Eq | Max Ineq | Mean Ineq |
| 512 | 1.5305 | 0.1041 | 0.01998 | 0.01285 | 0.000046 | 1.5904 | 0.1755 | 0.02403 | 0.01446 | 0.000060 |
| 1024 | 1.5097 | 0.08948 | 0.01872 | 0.01442 | 0.000049 | 1.5615 | 0.1201 | 0.02075 | 0.01892 | 0.000075 |
| 2048 | 1.4691 | 0.08005 | 0.01709 | 0.01170 | 0.000042 | 1.5534 | 0.08776 | 0.01839 | 0.01906 | 0.000073 |
| N | Sandwich BNN SvP, 10-Min | | | | | Supervised BNN SvP, 10-Min | | | | |
| | %Cost | Max Eq | Mean Eq | Max Ineq | Mean Ineq | %Cost | Max Eq | Mean Eq | Max Ineq | Mean Ineq |
| 512 | 1.5071 | 0.09438 | 0.01912 | 0.01453 | 0.000054 | 1.5807 | 0.1644 | 0.02323 | 0.01524 | 0.000061 |
| 1024 | 1.5194 | 0.08693 | 0.01865 | 0.01415 | 0.000047 | 1.5562 | 0.1108 | 0.02025 | 0.01679 | 0.000065 |
| 2048 | 1.5297 | 0.08532 | 0.01855 | 0.01288 | 0.000042 | 1.5387 | 0.08686 | 0.01843 | 0.01912 | 0.000072 |

*Table 7.* Comparison of Average Values (Over Five Trials) for BNN models: 10/15-min, Sandwich/Supervised. Note that this computation was done on a Max Mini M4 with 24GB RAM.

| N | Sandwich BNN, 15 Min | | | | | Supervised BNN, 15 Min | | | | |
|---|---|---|---|---|---|---|---|---|---|---|
| | %Cost | Max Eq | Mean Eq | Max Ineq | Mean Ineq | %Cost | Max Eq | Mean Eq | Max Ineq | Mean Ineq |
| 512 | 1.5298 | 0.1349 | 0.01810 | 0.01255 | 0.000045 | 1.5905 | 0.2281 | 0.02318 | 0.01425 | 0.000060 |
| 1024 | 1.5074 | 0.1106 | 0.01661 | 0.01407 | 0.000048 | 1.5594 | 0.1609 | 0.01957 | 0.01868 | 0.000074 |
| 2048 | 1.4687 | 0.09156 | 0.01474 | 0.01135 | 0.000041 | 1.5540 | 0.1114 | 0.01690 | 0.01890 | 0.000072 |
| N | Sandwich BNN, 10-Min | | | | | Supervised BNN, 10-Min | | | | |
| | %Cost | Max Eq | Mean Eq | Max Ineq | Mean Ineq | %Cost | Max Eq | Mean Eq | Max Ineq | Mean Ineq |
| 512 | 1.5054 | 0.1194 | 0.01710 | 0.01426 | 0.000053 | 1.5780 | 0.2204 | 0.02215 | 0.01502 | 0.000060 |
| 1024 | 1.5188 | 0.1064 | 0.01654 | 0.01376 | 0.000046 | 1.5573 | 0.1499 | 0.01888 | 0.01652 | 0.000064 |
| 2048 | 1.5284 | 0.09725 | 0.01623 | 0.01251 | 0.000041 | 1.5377 | 0.1081 | 0.01664 | 0.01887 | 0.000071 |

Figure 5 shows results of 5-Trials of proposed method of learning for three key metrics—% Gap, Max Eq., and Max Ineq.—across different sample sizes (N = 512, 1024, 2048) for 'case118' on a Mac Mini M4 with 24GB RAM. The red dashed lines with circular markers represent the mean values obtained from the sandwich training approach, while the blue solid lines with square markers correspond to the mean values from the supervised training approach. Error bars depict the minimum and maximum observed values for each method, providing insight into the variability of results. From the figure, we observe that as the number of samples increases, both methods exhibit a trend of increasing Max Eq. and Max Ineq. values, while % Gap remains relatively stable.

The supervised approach generally achieves lower variability (shorter error bars), indicating a more consistent performance. The sandwich method, while slightly less stable, maintains comparable mean values across the three metrics. These results suggest that supervised learning provides more predictable outcomes, whereas the sandwich method may introduce greater flexibility with minor variations in performance. Detailed results and and graphs are given in Table 7, Table 6, Figure 8 to Figure 15.

Results in Figure 8 to 15 are of five trials with identical hyperparameters demonstrate consistent improvement across all parameters except for max inequality. Max inequality remains challenging to suppress further, as it is already one order of magnitude better than max equality. Also the spread of error decreases with more samples indicating robust learning performance with more data. Note that these computations was done on a Mac Mini M4 with 24GB RAM.

### C.1. Larger System Results

*Table 8.* Comparative performance results for the ACOPF Problem for case500 with 512 labeled training samples, 2048 unlabeled samples and $T_{max} = 600$ sec.

| Method | Gap% | Max Eq. | Mean Eq. | Max Ineq. | Mean Ineq. |
|---|---|---|---|---|---|
| Sandwich BNN SvP (Ours) | 2.009 | **0.770** | 0.066 | 0.190 | **0.000** |
| Sandwich BNN (Ours) | 2.002 | 0.781 | **0.056** | 0.191 | 0.000 |
| Supervised BNN SvP (Ours) | 1.191 | 2.204 | 0.088 | 0.141 | 0.000 |
| Supervised BNN (Ours) | **1.191** | 2.401 | 0.072 | **0.140** | 0.000 |
| Sandwich DNN (Ours) | 1.782 | 13.800 | 0.268 | 0.2707 | 0.000 |
| Naïve MAE | 1.208 | 20.818 | 0.905 | 0.000 | 0.000 |
| Naïve MSE | 1.201 | 24.089 | 1.031 | 0.000 | 0.000 |
| MAE + Penalty | 1.205 | 11.833 | 0.580 | 0.000 | 0.000 |
| MSE + Penalty | 1.215 | 10.314 | 0.475 | 0.000 | 0.000 |
| LD + MAE | 1.279 | 11.166 | 0.532 | 0.000 | 0.000 |

*Table 9.* Comparative performance results for the ACOPF Problem for case2000 with 512 labeled training samples, 2048 unlabeled samples and $T_{max} = 600$ sec.

| Method | Gap% | Max Eq. | Mean Eq. | Max Ineq. | Mean Ineq. |
|---|---|---|---|---|---|
| Sandwich BNN SvP (Ours) | 0.514 | 5.114 | 0.324 | 0.196 | **0.000** |
| Sandwich BNN (Ours) | 0.503 | 5.409 | 0.262 | **0.187** | 0.000 |
| Supervised BNN SvP (Ours) | 0.461 | **4.107** | 0.238 | 0.917 | 0.000 |
| Supervised BNN (Ours) | **0.451** | 4.225 | 0.193 | 0.922 | 0.000 |
| Sandwich DNN (Ours) | 54.505 | 27.747 | 1.659 | 1.228 | 0.339 |
| Naïve MAE | 56.365 | 43.529 | 4.392 | 0.000 | 0.000 |
| Naïve MSE | 56.366 | 43.085 | 4.261 | 0.000 | 0.000 |
| MAE + Penalty | 56.349 | 19.591 | 1.055 | 0.000 | 0.000 |
| MSE + Penalty | 56.377 | 12.592 | 0.682 | 0.000 | 0.000 |
| LD + MAE | 57.257 | 18.870 | 0.647 | 0.000 | 0.000 |

Here, we present the complete results analogous to Tables 1 and 2 for the larger test cases: 'case500' and 'case2000' with 500 and 2000 nodes, respectively, in Tables 8 and 9. In general, it is clear from the tables that the approaches presented in this paper outperform the state-of-the-art DNN-based approaches when limited training time and compute resources are provided. Between the supervised and the sandwich BNN models, it appears that there is no clear winner. Table 8 indicates that the supervised BNN outperforms the sandwich BNN model on the Max. Ineq. gap metric for the 'case500' whereas the trend is reversed for the case2000' in Table 9. This minor variation is attributed to the fact that the maximum training time of $T_{max} = 600$ sec. is insufficient for both cases, and with more training time, these trends should look similar to the ones in Tables 1 and 2.

Another critical observation for both cases is that the 'Max. Eq.' value is substantially high even for the proposed best model (in Tables 8 and 9, respectively). However, it is clear that the proposed BNN proxies are better than standard DNN models with an order-of-magnitude difference. On the surface, readers may dismiss the efficacy of the proposed BNN models due to these large values. Still, these values have to be examined in conjunction with the error bounds on the voltage magnitude and voltage angles in Fig. 7. Examined together, the BNN model predictions have very low errors for the voltage magnitude and angles, and it is very much possible to develop a computationally inexpensive projection algorithm that projects this prediction onto the feasibility set of the ACOPF problem on lines of (Zamzam & Baker, 2020). This procedure would further reduce the Max. Eq. values and makes the projected solution usable. This is a minor detail regularly tackled in a power grid context and is not dealt with extensively in this paper.

Finally, Figure 6 shows the stabilization of the mean errors and the MPV with varying testing samples and posterior

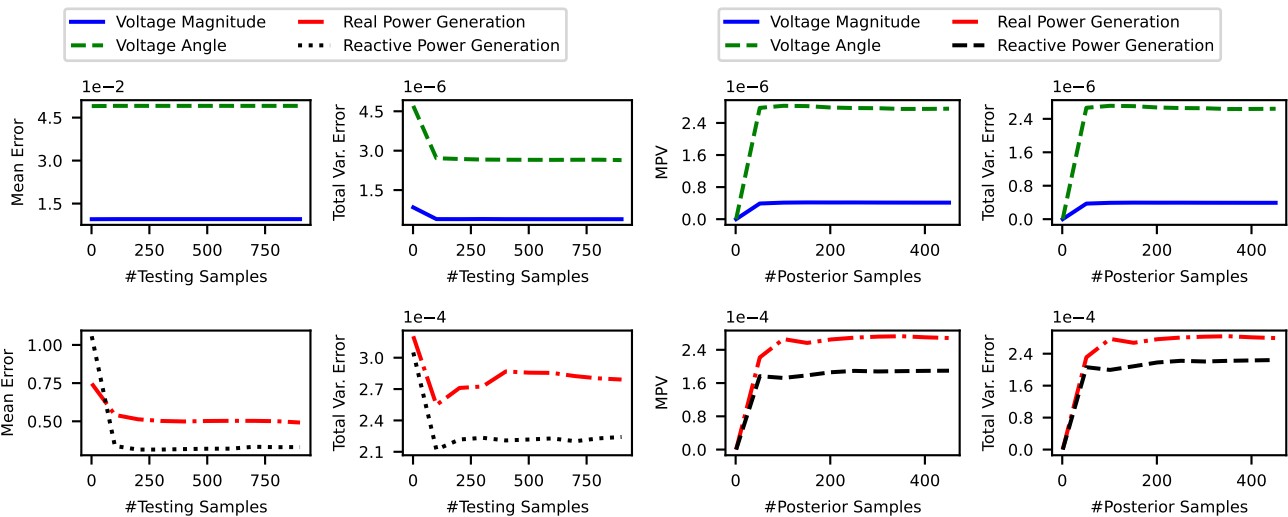

*Figure 6.* Convergence of mean error, MPV, total variance in error with respect to number of testing samples, and number of posterior samples.

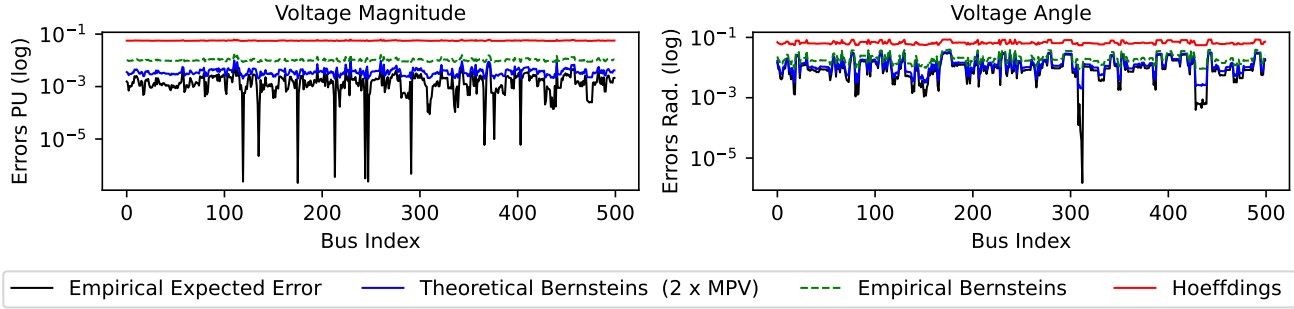

*Figure 7.* Comparison of voltage magnitude and voltage angle error bounds (in logarithmic scale) across bus indices for case500. The plot illustrates that PCBs using theoretical Bernstein bounds with $2 \times$ MPV from hypothesis (5) are tightest among all PCBs. We consider $\delta = 0.95$ and 1000 *out-of-sample* testing data points i.e. $M = 1000$.

samples, respectively. These plots validate the number of testing and posterior samples chosen for generating the results presented in Section 5.

## D. Performance on a non-OPF dataset

The semi-supervised training approach described in this work can be used to train optimization proxies for any constrained optimization problem over continuous variables. In this section we look at the performance of our method on the following non-convex optimization problem described in Donti et.al. (Donti et al., 2021).

$$\min_{y \in \mathbb{R}^n} \frac{1}{2} y^T Q y + p^T \sin(y), \quad \text{s.t. } Ay = x, \ Gy \le h. \tag{6}$$

The sinusoid function in the objective makes this a non-convex problem. We used the code provided by Donti et.al. to generate two sets of randomized instances of this problem. The comparative performance of the sandwiched training method against purely supervised BNN training is given in Table 10

| Model | (nV, nEq, nInEq) | Max-Eq violation | Max-InEq violation | Gap% | Time |
|---|---|---|---|---|---|
| BNN-Supervised | (70,20,50) | 0.484 | 0.170 | **2.6** | 800s |
| BNN-Sandwich | (70,20,50) | **0.298** | **0.109** | 7.1 | 800s |
| BNN-Supervised | (20,10,20) | 0.264 | 0.108 | 1.4 | 400s |
| BNN-Sandwich | (20,10,20) | **0.114** | **0.000** | **0.07** | 400s |

*Table 10.* nV, nEq and nInEq are respectively the number of variables, equality constraints and inequality constraints in the optimization problem. For these experiments we use $2^{12}$ samples in the supervised stage and $2^9$ samples in the unsupervised stage. Results above are evaluated on 100 testing instances and averaged over three different random instantiations of the training procedure. These show the advantages of using the sandwiched approach for training a BNN model in the low-data/time-constrained regime for this problem. . The hardware used here is the Google Collab instance with Intel(R) Xeon(R) CPU @ 2.20GHz.

## E. Concentration Bounds

This section presents three bounds used in Section 4. Here, $X$ represents a generic random variable and is not related to the optimization proxy variables. Since these are well-known inequalities, we omit the proofs, which can be found in the respective references.

**Theorem E.1** (Hoeffding's). *(Hoeffding, 1994) Let $X_1, \ldots, X_M$ i.i.d. random variables and suppose that $|X_i| \leqslant R$ with expectation $\mathbb{E}(X_i)$, and let $\bar{X}_M = \frac{1}{M} \sum_{i=1}^{M} X_i$. Then, with probability at least $1 - \delta$,*

$$\left| \bar{X}_M - \mathbb{E}(X_i) \right| \leqslant R \sqrt{\frac{\log(2/\delta)}{2M}}.$$

.

**Theorem E.2** (Empirical Bernstein). *(Audibert et al., 2007; Mnih et al., 2008) Let $X_1, \ldots, X_M$ be i.i.d. and suppose that $|X_i| \leqslant R$, expectation $\mathbb{E}(X_i)$ and let $\bar{X}_M = \frac{1}{M} \sum_{i=1}^{M} X_i$. With probability at least $1 - \delta$,*

$$\left| \bar{X}_M - \mathbb{E}(X_i) \right| \leqslant \sqrt{\frac{2 \widehat{\mathbb{V}} \log(3/\delta)}{M}} + \frac{2R \log(3/\delta)}{M}$$

*where, $\widehat{\mathbb{V}} = (1/M) \sum_{i=1}^{M} (X_i - \bar{X}_M)^2$ is empirical variance.*

**Theorem E.3** (Bernstein). *(Sridharan, 2002) Let $X_1, \ldots, X_M$ be i.i.d. and suppose that $|X_i| \leqslant R$, mean $\mathbb{E}(X_i)$ and $\mathbb{V} = \text{Var}(X_i)$. With probability at least $1 - \delta$,*

$$\left| \bar{X}_M - \mathbb{E}(X_i) \right| \leqslant \sqrt{\frac{2 \mathbb{V} \log(1/\delta)}{M}} + \frac{2R \log(1/\delta)}{3M}$$

*Table 11.* Error bound $\varepsilon$ in PCBs, provided by different concentration inequalities.

| Hoeffding's | Empirical Bernstein | Bernstein |
|---|---|---|
| $R\sqrt{\frac{\log(2/\delta)}{2M}}$ | $\sqrt{\frac{2\widehat{\mathbb{V}}_e \log(3/\delta)}{M}} + \frac{3R \log(3/\delta)}{M}$ | $\sqrt{\frac{2(2 \times \text{MPV}) \log(1/\delta)}{M}} + \frac{2R \log(1/\delta)}{3M}$ |

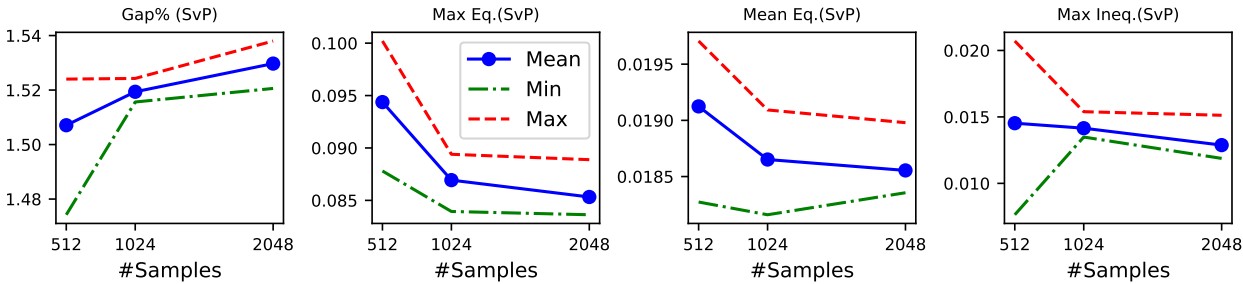

*Figure 8.* Performance metrics for Sandwich BNN SvP model with 10-minute training.

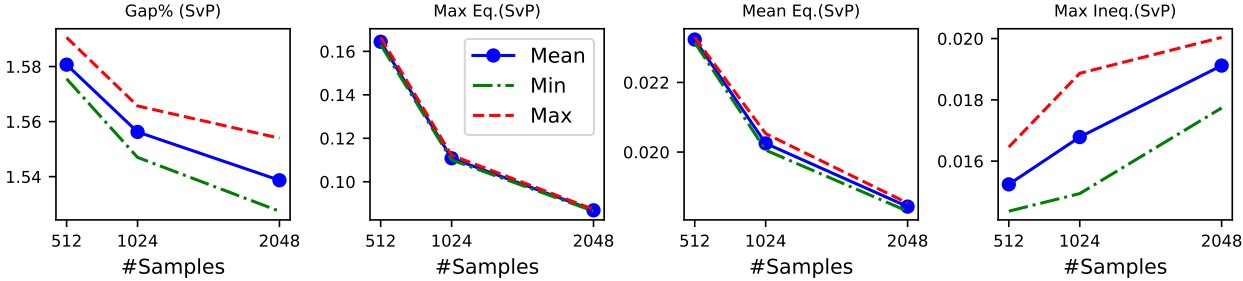

*Figure 9.* Performance metrics for Supervised BNN SvP model with 10-minute training.

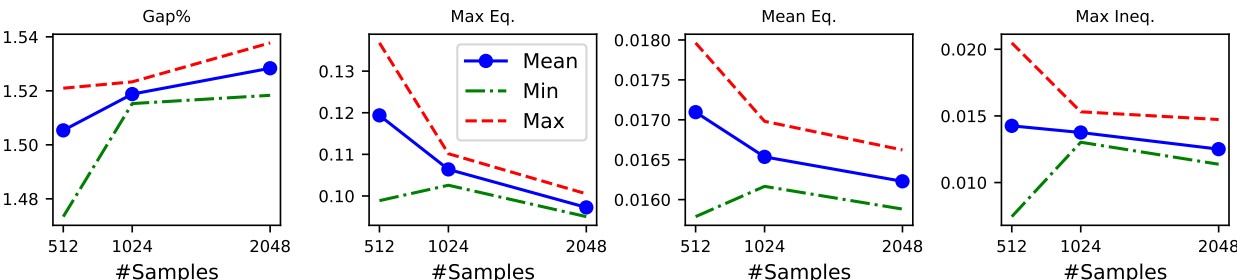

*Figure 10.* Performance metrics for Sandwich BNN model with 10-minute training.

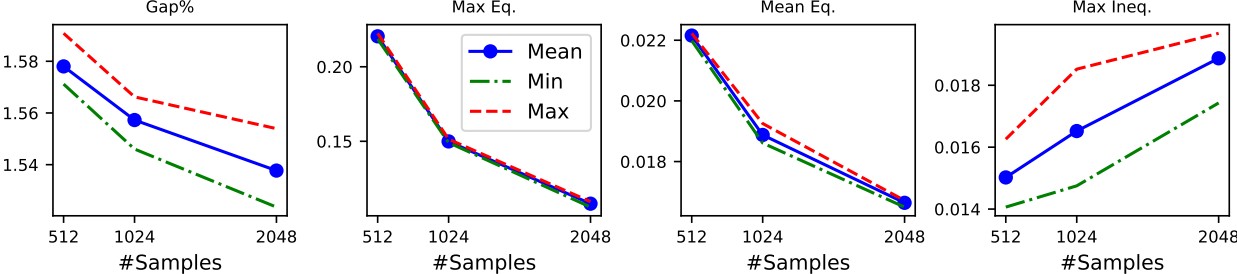

*Figure 11.* Performance metrics for Supervised BNN model with 10-minute training.

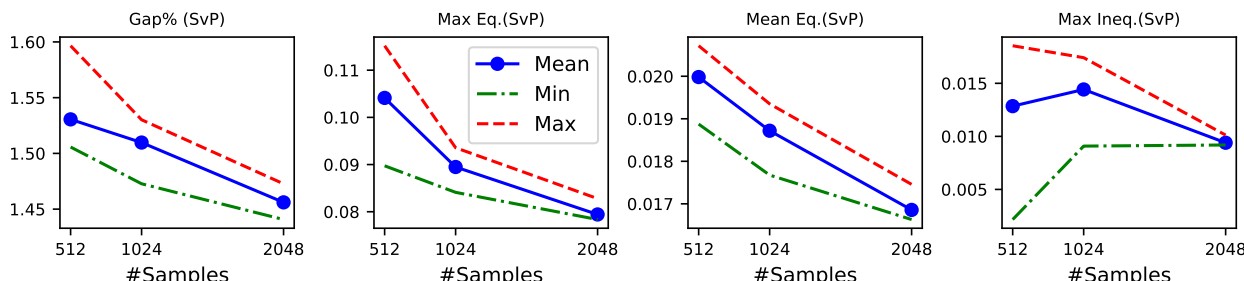

*Figure 12.* Performance metrics for 'Case118' with fixed unsupervised samples (2048) over $T_{max} = 900sec$ (15 minutes) using the Sandwich BNN SvP model.

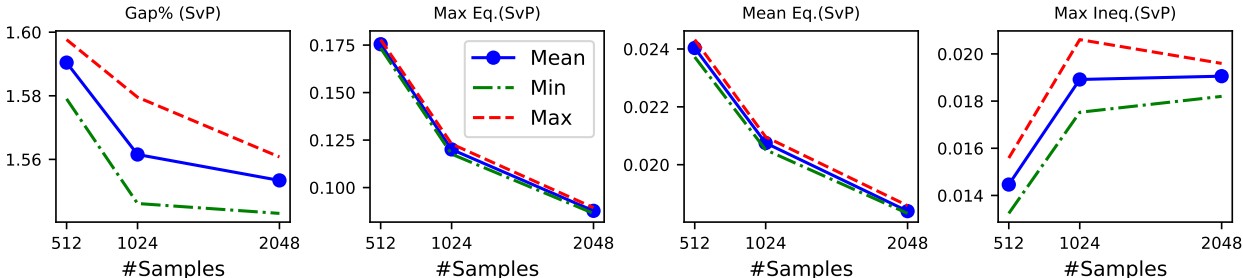

*Figure 13.* Performance metrics for Supervised BNN SvP model model with 15-minute training.

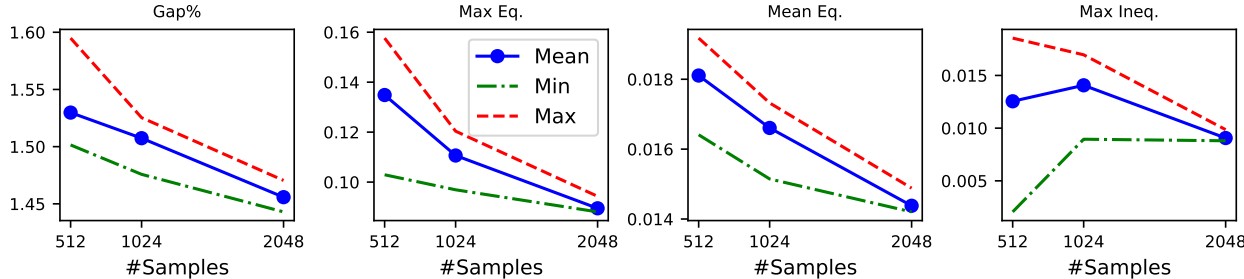

*Figure 14.* Performance metrics for Sandwich BNN model with 15-minute training.

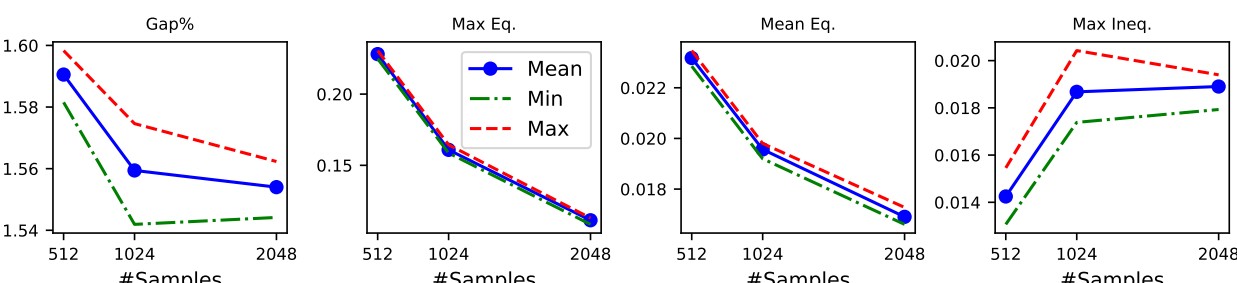

*Figure 15.* Performance metrics for Supervised BNN model with 15-minute training.

