# OpenReview forum: "Optimization Proxies using Limited Labeled Data and Training Time -- A Semi-Supervised Bayesian Neural Network Approach"
_ICML.cc/2025/Conference — ICML 2025 poster_

### Official Review · Reviewer_2Rny · 2025-03-15

**Overall Recommendation:** 3

**Summary:**

This paper proposes and evaluates a semi-supervised approach to training Bayesian Neural Network (BNN) optimization proxies for predicting solutions to constrained optimization problems. Specifically, the paper proposes augmenting training on labeled data (from solving optimization problem instances) with training on unlabeled data about constraint satisfaction in a "sandwich" fashion. The paper is concerned with settings in which both labeled data (for training and validation) and training time for the model are constrained. The authors argue that the use of BNNs allows both an increase in sample efficiency compared to (non-Bayesian) DNNs as well as a principled predictive uncertainty quantifications, for which they develop a new approach based on Bernstein concentration bounds. The paper evaluates the proposed approach on optimal electric power flow optimization benchmarks, demonstrating improved performance in the small-sample regime compared to a number of baseline methods.

### Update after rebuttal
Overall, given the authors' responses I feel slightly more positive about the paper, but not enough to raise my score to a full accept. For that I would still like to see a more detailed ablation study on the training setup, as well as a comparison with more compute-intensive baselines (even though they may not be feasible from an applied perspective).

**Claims And Evidence:**

The claims appear well-supported overall. However, the empirical evaluation could be more comprehensive and include some other (e.g. non-power-flow) example applications as well as more detailed ablation studies of some of the hyperparameters (e.g. number of unlabeled examples, learning rates, etc.)

**Essential References Not Discussed:**

I am not familiar enough with the specific literature to assess this properly.

**Experimental Designs Or Analyses:**

Not in detail, the experimental setup seems reasonable (modulo the comments on evaluation criteria above).

One thing I'd like to see more of are ablations of some of the training settings. BNN learning can be rather finicky, and so it would be important to understand how sensitive the setup is to the number of training samples and unlabeled samples and hyperparameters such as learning rates etc.

**Methods And Evaluation Criteria:**

* The optimal power flow problems are fine benchmark problems, but given that the proposed method is a generic one I would have expected to see some problems from other applications to better understand how the approach performs in other domains.
* The evaluation criteria generally make sense to me - as someone with limited familiarity with the specific literature in this area. However, I am surprised to only see the performance compared on single instances without providing variances across the randomness in the results (e.g. w.r.t. the testing instances / data generation / splitting and w.r.t. to the weight initialization and other randomness in the methods themselves). This leaves open the possibility that the provided results are somewhat cherry-picked -- I am not suggesting that they are, rather that it's not clear to me from the paper, and the paper could to better at proactively assuaging this concern.
* The evaluation excluded a number of baseline methods that would be interesting to have for comparison, such as self-supervised constrained optimization methods and Graph Neural Network-based large models. The authors rightfully argue that these methods may not be feasible in the setting targeted by the paper due to excessive computational requirements -- however, it would be very helpful from a scientific perspective to see the performance penalty incurred by using fewer samples with the proposed approach to understand the broader tradeoffs between the different methods.
* Related to the above, the paper also mentions Ensemble DNNs as an alternative approach, but does not discuss them in detail due to "very high computational requirements". It's not necessarily clear to me that this will always be an issue. For instance, in settings where label generation is extremely costly but DNN training time is not a major concern, ensemble DNNs could be a good alternative. Given that a claimed primary contribution is the uncertainty quantification, having the evaluation against ensemble DNNs as at least one baseline would be highly valuable.

**Other Comments Or Suggestions:**

N/A

**Other Strengths And Weaknesses:**

* The paper is generally well written. I appreciated that it does a decent job introducing the problem setting and some of the background material, allowing me as someone less intimately familiar with the literature to follow the presentation without too much trouble in most places.
* The focus on limited data and training time and comparing performance under these constraints is a nice way of looking at the practicality of the approach (though I would like to understand the gap to an ideal baseline, see comments above)

**Questions For Authors:**

* You mention that "the ‘Max Ineq.’ growth could be easily suppressed by incorporating bound repair layers, as used in DNN models in ML4OPF. Why not do this? This would be a very useful ablation to understand the effect of a bound repair layer on the performance. Especially in light of the fact that you observe that "Sandwich BNN SvP shows slightly higher scaling inequality gaps".
* It appears that Sandwich learning is a lot more helpful for case57 - why? Is this b/c constraints are more challenging to satisfy in this problem? It would be helpful if you could provide this kind of intuition more broadly in the discussion of your results.
* How to select the number of unlabeled examples and schedule the learning steps such that the model doesn't over-fit to the constraints? Is there a principled way to do this? Can you provide any ablations on this?

**Relation To Broader Scientific Literature:**

* The main novel angle of the paper is the focus on and analysis of the small-sample regime in which labeled data and training time is limited. Previous works have often focused on settings where that data and time was less of a constraint, so the paper provides an interesting additional perspective.
* Both the use of BNNs as optimization proxies and the semi-supervised data augmentation using feasibility information appears novel (though I have very limited familiarity with the literature).

**Theoretical Claims:**

N/A -- there are no novel theoretical claims (the authors apply a "theoretical" version of Bernstein's inequality using the Mean Predictive Variance, but are basing this on a hypothesis that they only check empirically in a limited number of examples).

---

> ### Author Rebuttal · Authors · 2025-04-01
>
> **Methods and evaluation criteria**
>
> We agree with the reviewer that BNN training can be sensitive to various random choices in the training. We have run a new batch of experiments for the 118 bus problem with increased dataset size. The variance in the results from different learning experiments can be seen here. The time budget used here is 900s. Results are slightly different from what is reported in the paper due to us using a different machine. Despite these changes, we see that the results are quire robust.
>
> #### **Comparison of Max, Avg, and Min values for different $N$ for 15 Min Training on `case118`**
>
> | N    | Type | %Gap  | Max Eq  | Max Ineq |
> |------|------|-------|--------|---------|
> |      | Max  | 1.597 | 0.115  | 0.019   |
> |  512  | Avg  | 1.531 | 0.104  | 0.013   |
> |      | Min  | 1.506 | 0.090  | 0.002   |
> |-----------|
> |      | Max  | 1.530 | 0.094  | 0.017   |
> |  1024 | Avg  | 1.510 | 0.089  | 0.014   |
> |      | Min  | 1.473 | 0.084  | 0.009   |
> |---------|
> |      | Max  | 1.473 | 0.083  | 0.010   |
> |  2048 | Avg  | 1.456 | 0.079  | 0.009   |
> |      | Min  | 1.441 | 0.078  | 0.009   |
>
> More experiment results can be found here: [Link](https://drive.google.com/file/d/1sm9eEiYutFk89ZqKwJNI4pf8HQueUJmy/view?usp=share_link)
>
> **Comparison with ensemble DNN and GNN**
>
> We agree that ensemble methods can potentially help in quantifying uncertainty. Our comparison, for the OPF benchmark, is currently with state of the art scalable Neural network based methods given our limited training time and data availability. At present we are not aware of any existing work on ensemble methods for the case of OPF. If individual ML methods are trained sequentially without our framework, scalability might be an issue. However developing ensemble methods for OPF and comparing with our BNN model will an useful avenue of future work.
>
> **Questions**
>
> **1.** We did not do this for our experiments as we see that the inequality violations are already low, especially in comparison to the equality violations. Moreover, our mean inequality Gap results are similar to that of DNNs with repair obtained by ML4OPF tools.  Due to limited time for rebuttable we did not setup those bound repair experiments with BNN. Also, note that criteria of SvP selection is only minimization of maximum equality gap. We also note that one can include inequality constraints in SvP criteria to improve upon further. We can include it in our final draft.
>
> **2.** This effectiveness of sandwich method for case57 likely due to the limited time budget of 600s that we have used for all the experiments in the paper. The sandwich BNN for case57 has ample time to converge, but larger cases are stopped early. Even under this constraint, we see that the Sandwich BNN still outperforms other methods even for larger cases. Further, case57 has less active constraints and thus intuitively we can say that unsupervised data is able to enforce feasibility much better in sandwiched learning.
>
> We conduct a robustness study on `case118' by running five trials with training times of 10 and 15 minutes, using 512, 1024, and 2048 supervised training samples, while keeping 2048 unsupervised samples constant across all models and trials. The results clearly demonstrate that increasing training time and the number of supervised samples consistently reduces \%Gap, Max Eq., and Max Ineq. Additionally, Mean Eq. remains an order of magnitude lower, and Mean Ineq. stays at the order of $10^{-5}$. For instance, the Sandwich BNN SvP model achieves a 1.50\% Gap, 0.094 Max Eq., and 0.014 Max Ineq. with 512 supervised samples and a 10-minute training time, averaged over five trials. In contrast, with 2048 supervised samples and 15 minutes of training, the same model improves to a 1.46\% Gap, 0.080 Max Eq., and 0.011 Max Ineq. This implies that increasing training time by five minutes while quadrupling the supervised samples results in a 2.66\% reduction in optimality error (\%Gap), a 17.5\% reduction in equality constraint feasibility error (Max Eq.), and more than a 27\% reduction in inequality constraint feasibility error (Max Ineq.).
>
> **3.** We find that early stopping the unsupervised round is necessary to prevent it from overfitting. The intuition we follow is that we do not want to fit either supervised  or unsupervised models too well, to avoid biasing our surrogate towards either feasibility or optimality. Further, we observe that too many unlabeled examples will require too much compute to extract any meaningful information, thus we kept them constant at 2048. We fix each round of unsupervised training to 150 sec. and supervised training time as 100 sec. for 512 samples. While we do not fine tune this choice in our experiments, cros-validation on different relative training sizes might be a way to understand the dependence of hyperparameters that includes number of unlabeled examples and division of time between supervised and unsupervised training.

---

> > ### Comment · Reviewer_2Rny · 2025-04-04
> >
> > Thanks for the the clarifications and additional results.
> >
> > The ablations in terms of the number of samples and training time are helpful. When talking about robustness I was also concerned with things like learning rates etc. I don't think these ablations are crucial, but it would be helpful to understand how sensitive the results are to changes to the training setup.
> >
> > > The optimal power flow problems are fine benchmark problems, but given that the proposed method is a generic one I would have expected to see some problems from other applications to better understand how the approach performs in other domains.
> >
> > This concern of mine still stands - The method is presented as a generic method, but evaluation is only done on OPF problems. Having results on at least one other problem domain would make me more confident in the general applicability of the results.
> >
> > > The evaluation excluded a number of baseline methods that would be interesting to have for comparison, such as self-supervised constrained optimization methods and Graph Neural Network-based large models. The authors rightfully argue that these methods may not be feasible in the setting targeted by the paper due to excessive computational requirements -- however, it would be very helpful from a scientific perspective to see the performance penalty incurred by using fewer samples with the proposed approach to understand the broader tradeoffs between the different methods.
> >
> > These kinds of comparisons would still be helpful, at least to have a ballpark understanding of how well the proposed approach works compared to other (possibly infeasible) benchmarks.

---

> > > ### Author Response · Authors · 2025-04-09
> > >
> > > The referee is right in pointing out that our method is demonstrated for only OPF, although the method it self is presented for a general problem. To address this,we have modified our code to deal with general constraints optimization problems and have run benchmarks on the following optimization problem also used by Donti et.al [1] as a non-convex benchmark,
> > >
> > > $\min_{y \in \mathbb{R}^{n}} \quad  \frac12 y^{T} Qy + p^{T}\sin(y),$
> > > $~\text{s.t.} ~Ay = x $
> > >                 $,~~ Gy \leq h.$
> > >
> > > We study the performance of the sandwich BNN method and the standard supervised BNN with only 512 supervised samples on this problem. For the sandwich method, we use 8x unsupervised samples in the unsupervised layers. Preliminary results of this are given below for two data sets generated from the above problem with 20 and 70 variables respectively. The data generation procedure is as given in [1].
> > >
> > > | Model            |(nV,nEq,nInEq)| Gap%  | Max Eq violation | Max InEq violation |$T_{max}$   |
> > > |------------------|--------------|-------|------------------|--------------------|----------|
> > > | BNN (supervised) | (20,10,20)   | 2.32  | 0.5092           | 0.262              |400       |
> > > | BNN (sandwich)   | (20,10,20)   | 0.71  | 0.140            | 0.001              |400       |
> > > | BNN (supervised) | (70,20,50)   | 6.48  | 1.711            | 0.253              |800       |
> > > | BNN (sandwich)   | (70,20,50)   | 4.84  | 1.862            | 0.213              |800       |
> > >
> > > nV, nEq and nInEq are respectively the number of variables, equality constraints and inequality constraints. Results above are evaluated on 100 testing instances. These show the advantages of using the sandwiched approach for training a BNN model in the low-data/time-constrained regime for this problem. A more comprehensive version of this experiment can be added to the camera ready version of the manuscript.
> > >
> > > [1] Donti, Priya L., David Rolnick, and J. Zico Kolter. "DC3: A learning method for optimization with hard constraints." ICLR (2021).

---

### Official Review · Reviewer_iXWs · 2025-03-18

**Overall Recommendation:** 3

**Summary:**

The paper introduces a Bayesian Neural Network (BNN) framework as a surrogate model for constrained optimization, leveraging its intrinsic uncertainty quantification for robust predictions. It employs a semi-supervised training strategy that alternates between supervised learning on limited labeled data and unsupervised learning that enforces constraint feasibility via data augmentation. Experiments  show that this approach significantly reduces constraint violations and prediction errors, outperforming traditional deep neural network methods in low-data, low-compute settings. Additionally, by applying Bernstein’s inequality, the model derives tight probabilistic confidence bounds, offering practical error estimates without extensive validation data.

**Claims And Evidence:**

The experimental results in the submission strongly support its core claims within the context of AC optimal power flow problems, demonstrating improvements in constraint satisfaction and reduced prediction errors under low-data and low-compute conditions. However, the generality of the approach beyond ACOPF is less clearly supported, as most evidence is specific to this application domain. In addition, the claim that a simple multiplier (2×MPV) reliably bounds the total variance in error is based on a hypothesis validated only in these experiments and may require further theoretical backing and broader empirical evaluation. Overall, while the claims are convincing for the studied case, their applicability to other constrained optimization problems remains somewhat problematic.

**Essential References Not Discussed:**

Yes, there are related works that could further contextualize the paper’s key contributions, particularly in the areas of surrogate modeling with limited data and constraint handling. For example:
- ***Prior Fitted Neural Network as Surrogate:***
Müller, Samuel, Matthias Feurer, Noah Hollmann, and Frank Hutter’s “Pfns4bo: In-context learning for Bayesian optimization” (ICML 2023) addresses the challenge of limited data by leveraging a prior fitting approach. This work provides an alternative perspective on building surrogate models that effectively operate in data-scarce regimes, which is closely related to the paper’s focus on constrained optimization proxies.

- ***Boundary Exploration for Bayesian Optimization with Unknown Physical Constraints:***
Tian, Yunsheng, Ane Zuniga, Xinwei Zhang, Johannes P. Dürholt, Payel Das, Jie Chen, Wojciech Matusik, and Mina Konakovic Lukovic’s work “Boundary Exploration for Bayesian Optimization With Unknown Physical Constraints” (ICML 2024) employs ensemble methods to explore constraint boundaries. This study offers insights into handling unknown physical constraints through an ensemble approach, complementing the paper’s sandwich training framework that enforces feasibility via semi-supervised learning.

Including and discussing these works would enhance the understanding of how the paper’s contributions fit within the broader landscape of surrogate-based optimization and constrained learning methods.

**Experimental Designs Or Analyses:**

I examined the experimental designs used for evaluating the ACOPF problems across different system sizes, where the authors compare metrics such as optimality gap, equality/inequality violations, and error bounds among various models. The overall design is sound for the application at hand, with a reasonable selection of benchmark datasets and a clear focus on both supervised and semi-supervised BNN approaches. However, one issue is that the analyses do not include any confidence intervals or error bars for the reported metrics, which makes it harder to assess the statistical significance and variability of the improvements. This omission limits our ability to fully evaluate the robustness of the experimental findings.

**Methods And Evaluation Criteria:**

Yes, the methods and evaluation criteria are well-aligned with the application. The paper’s use of semi-supervised Bayesian neural networks addresses the challenge of limited labeled data and short training times, which are key constraints in real-world power grid optimization. Evaluation metrics such as the optimality gap, equality/inequality constraint violations, and probabilistic confidence bounds directly assess performance in areas critical to ACOPF problems. Moreover, using established benchmark datasets ensures that the evaluation is robust and comparable to baselines.

**Other Comments Or Suggestions:**

Merged with the section below.

**Other Strengths And Weaknesses:**

Strengths:
- The paper introduces a novel sandwich training framework that alternates between supervised and unsupervised phases to enforce constraint feasibility. Importantly, this framework is not limited to Bayesian Neural Networks (BNNs), suggesting broader applicability to various neural architectures in surrogate modeling and optimization tasks.

Weaknesses:
 - The focus on surrogate modeling involves a largely passive approach to identifying feasibility, actively optimizing only within a pre-identified feasible region rather than integrating feasibility and optimality exploration simultaneously. This approach is reminiscent of recent work such as CONFIG by Xu et al. (Xu, Wenjie, Yuning Jiang, Bratislav Svetozarevic, and Colin Jones. “Constrained efficient global optimization of expensive black-box functions.” In International Conference on Machine Learning, pp. 38485-38498. PMLR, 2023), which more actively integrates constraint exploration. Additionally, the paper does not provide any regret guarantees, a limitation also noted in similar treatments like CONFIG.

**Questions For Authors:**

1.  While the sandwich training framework is demonstrated with BNNs, have you considered or experimented with applying it to other neural architectures, and what challenges do you anticipate in such extensions?

2. The experiments currently lack confidence intervals or statistical significance tests. Could you provide additional details on the variability of your results and any measures taken to assess statistical robustness?

3.  Given that similar approaches like CONFIG (Xu et al., 2023) offer integrated exploration and regret guarantees, do you see a pathway for incorporating regret guarantees into your framework, and how might that impact performance?

**Relation To Broader Scientific Literature:**

The paper’s key contributions are well-rooted in existing literature on constrained optimization and semi-supervised learning. Notably, the authors propose a sandwich training framework that alternates between a supervised phase on limited labeled data and an unsupervised phase that enforces constraint feasibility via data augmentation. Although demonstrated with Bayesian Neural Networks (BNNs) to leverage uncertainty quantification and tight error bounds via Bernstein’s inequality, the sandwich training framework itself is not limited to BNNs; it has broader potential for application across various neural architectures. This approach builds on prior methods—such as penalty methods and projection techniques for constrained optimization, as well as semi-supervised learning strategies like pseudo-labeling—and extends these ideas to effectively address real-world problems with scarce labeled data and strict operational constraints.

**Theoretical Claims:**

I reviewed the derivations related to the probabilistic confidence bounds, which rely on established concentration inequalities like Hoeffding’s and Bernstein’s inequalities. The proofs themselves are adaptations of well-known results rather than new theoretical contributions, and they primarily serve to justify the use of the 2×MPV heuristic as a bound on the total variance in error. I did not encounter any significant issues in the correctness of these derivations, but it’s important to note that the paper does not present any substantial new theory. Overall, the theoretical claims are based on solid, previously established results, with the novel contribution being in their application within the BNN-based optimization proxy framework.

---

> ### Author Rebuttal · Authors · 2025-04-01
>
> **Claims And Evidence: The 2×MPV Heuristic and Its Broader Validation**
>
> We appreciate this insightful comment. We believe that the 2MPV heuristic is sufficient to capture total variance in the error, as substantiated by the studies shown in Fig 4. This heuristic will require similar empirical validation before it can be used for other datasets. While the law of total variances argument in Eq.(5) makes the first step in understanding the theory behind this, a full theoretical understanding of why one term dominates at selected training data sizes will be the object of a future work.
>
> **Experimental Designs Or Analyses: Lack of Confidence Intervals or Statistical Significance Tests**
>
> We acknowledge the importance of assessing the robustness of our experimental results. We have run a new batch of experiments for the 118 bus problem with increased dataset size. The variance in the results from different learning experiments can be seen here. The time budget used here is 900s or 15 mins. Results (including the base case) are slightly different from what is reported in the paper due to the use of a different machine. Despite these changes, we see that the results are quire robust.
>
> #### **Comparison of Max, Avg, and Min values for different $N$ for 15 Min Training on `case118`**
>
> | N    | Type | %Gap     | Max Eq   | Max Ineq  |
> |------|------|----------|---------|----------|
> | 512  | Max  | 1.59 | 0.115 | 0.018  |
> | 512  | Avg  | 1.53  | 0.104 | 0.012 |
> | 512  | Min  | 1.50 | 0.089 | 0.002 |
> |-|
> | 1024 | Max  | 1.53 | 0.093 | 0.017 |
> | 1024 | Avg  | 1.50  | 0.089 | 0.014  |
> | 1024 | Min  | 1.47 | 0.084  | 0.009 |
> |-|
> | 2048 | Max  | 1.47 | 0.082 | 0.010 |
> | 2048 | Avg  | 1.45 | 0.079 | 0.009 |
> | 2048 | Min  | 1.44 | 0.078 | 0.009 |
>
>
> Updated results with error bars can be found here: [Link](https://drive.google.com/file/d/1sm9eEiYutFk89ZqKwJNI4pf8HQueUJmy/view?usp=share_link)
>
> **Essential References Not Discussed:**
>
> We appreciate the reviewers' suggested works. While related, we note that our focus is not directly on "Bayesian optimization" but on building a "Bayesian surrogate" for the optimization problem. In particular, we do not focus on selecting additional training data via acquisition functions/active learning. However, techniques mentioned in (paper 1) such as PFNs can help potentially reduce training time of Bayesian inference used within our framework (as shown in *Transformers Can Do Bayesian Inference*, ICLR 2022). However, PFNs have so far been applied only to relatively small BNNs due to the high computational cost of the attention mechanism. We are actively exploring ways to scale PFNs, such as using linear attention or alternative architectures like CNNs.
>
> Unlike boundary exploration for identifying unknown constraints (paper 2), the constraints are explicitly defined in our work. It will be of interest to integrate our approach with boundary exploration—potentially for feasibility verification or active learning.  We will include a discussion of these works to better position our contributions.
>
> **Weaknesses:**
>
> We agree that actively exploring feasibility and optimality is an important direction for future research. Our problem is not black-box optimization but rather surrogate modeling. It is challenging to model optimality in BNN with unsupervised data, as the target distribution of optimality is not known a priori without supervised data. Only the feasibility target distribution can be determined, which corresponds to a delta distribution at zero. Therefore, the simultaneous exploration of both optimality-and feasibility, including the right mix of training data with sandwich approach, will be our future work.
>
> **Questions**
>
>  1. As shown in the original draft, we did sandwich learning on DNN and showed that under low supervised data, BNN outperforms DNN considerably.  Also, the comparative results with other DNN methods indicate that sandwiching remains effective.
>
> 2.  As shown in (min-max) Table above,  we see that our results are quite robust to changes in the training set, time budget and report the max and min values of all the performance metrics and additional results in an updated pdf file here [Link](https://drive.google.com/file/d/1sm9eEiYutFk89ZqKwJNI4pf8HQueUJmy/view?usp=share_link)
>
> 3. The current work is not focused on active learning or intelligent sampling of training data. Incorporating regret guarantees is an interesting direction for future work. One possible direction we see is to use a sandwich model with very low supervised data, which can provide a computationally cheap baseline, then use that as a surrogate in active search of training samples and developing regret guarantees.

---

### Official Review · Reviewer_Uqx1 · 2025-03-20

**Overall Recommendation:** 3

**Summary:**

The paper proposes a new Bayesian Neural Network (BNN) for solving non-linear constrained optimization problems that can be computationally expensive. The Bayesian Neural Network serves as a proxy for solving the original problem and is computationally more efficient. The downside of similar existing approaches is that they can perform poorly if labeled data and training times are limited. To address this deficiency, the paper's BNN leverages unsupervised learning that enforces constraint feasibility. Specifically, the model alternates between phases of supervised and unsupervised learning. Finally, it samples weights for the neural network from the posterior distribution to generate a posterior prediction matrix and selects the weight that best satisfies the equality constraints. The paper then demonstrates the effectiveness of their approach compared to deep neural network alternatives against standard benchmarks.

**Claims And Evidence:**

The work is mainly empirical and the claims are supported by the numerics seem to make sense.

**Essential References Not Discussed:**

I am less familiar with this area and thus do not have any additional reference suggestions.

**Experimental Designs Or Analyses:**

I primarily evaluated the experimental design of the paper which seems to use standard benchmarks from the open-source OPFDataset from Torch Geometric. The results seem to make sense and seem promising. The only issues I see are the robustness of the results as they only focus on testing their approach on a single sample size and training time setting. It would be interesting to see how robust their results are as you vary sample size and training time, especially since the paper claims their approaches are computationally efficient.

**Methods And Evaluation Criteria:**

The proposed method and evaluation criteria make sense for the problem and application. The paper utilizes unsupervised learning to address limited data and a Bayesian approach to search for solutions that are more feasible. While I am less familiar with the evaluation criteria, it seems to be a standard benchmark for this type of work.

**Other Comments Or Suggestions:**

1. Typo on pg 2:  "10 minutes of training tim on a single CPU core"
2. In Eq (3) you give the constraint $g$ an index $I$. It may be more clear if you included it in the formulation of Eq (1).
3. Missing a word: "Sandwich DNN in these tables a DNN with the same network architecture as the BNN, trained under the same time constraints."

**Other Strengths And Weaknesses:**

Strengths
1) The proposed approach produces compelling numerical results in low-data settings, which for the most part make intuitive sense or are discussed by the authors.
2) The choice to combine Bayesian neural networks and unsupervised learning makes intuitive sense as both are suitable for low-data settings. Being able to sample multiple predictions is also an interesting feature as it allows users to cheaply tune the level of feasibility of the solutions which may be useful in practice.
3) The paper generally feels well written and well organized.

Weaknesses
1) The paper only focuses in low-data and low-computation settings so for readers less familiar with the literature it's hard to contextualize how the method performs compared to more computationally expensive approaches or approaches that do better with more data. They also do not scale the amount of data so it's hard to understand if their approach also performs well when the amount of data increases or how the improvement scales as data increases.
2) The improved performance of the approach is not uniform overall the experiments, specifically the ones found in the appendix. The paper makes some claims on why (mainly related to training time), so it would be useful to numerically verify the claims.
3) There are some minor clarity issues related to notation. For example, it was hard to understand how exactly $\mathcal{D}\_f$ was constructed based on the notation and the notation $p_{W}^{m}$ doesn't seem to be explicitly defined in the main body.

**Questions For Authors:**

1) I don’t understand how $\mathcal{D}\_{f}$ is constructed since the notation $\mathcal{D}\_f = \left( (\bf{x}\_j, \mathcal{F}(\cdot, x) \right)^{M}_\{j=1}$ is confusing. Are you trying to say you just sample some arbitrary feasible solution? Why is that computational inexpensive? Can more detail be provided?
2) What is $p_{W}^{m}$? It should maybe be formally defined.
3) When you alternate between unsupervised and supervised learning, is there benefit in learning until parameters converge?
4) To clarify, is the output of the model $f_{W^*}(\bf{x}^t)$?
5) Does performance improve with more unsupervised data?
6) How does performance scale as you increase data?
7) Are there any hyper parameters that need to be tuned for the BNN? How does the tuning affect performance?
8) How do you choose the variance parameters for the prior and likelihood? How do these parameters affect the performance of your model?

**Relation To Broader Scientific Literature:**

The key contributions of the paper build on the literature of end-to-end constrained optimization learning which is well outlined in the survey paper [1]. Specifically, the paper focuses on machine learning predicting solutions to constrained optimization problems. Within the space, they focus on the key challenges such as limited data and expensive compute during training.

___
[1] Kotary, J., Fioretto, F., van Hentenryck, P., and Wilder, B. End-to-end constrained optimization learning: A survey. In 30th International Joint Conference on Artificial Intelligence, IJCAI 2021, pp. 4475–4482. International Joint Conferences on Artificial Intelligence, 2021.

**Theoretical Claims:**

The paper makes few theoretical claims, mainly those related to Bernstein and Hoeffding bounds which seem correct.

---

> ### Author Rebuttal · Authors · 2025-04-01
>
> **Weaknesses**
>
> **1. Scalability**
> We agree that exploring scalability is important. To address this concern, we have added additional experimental results showing that while our method excels in low-data regimes, it also remains competitive as the amount of available data increases. These experiments are performed on the case118  system due to the limited rebuttal window. But similar experiments can be performed and added to the paper for the camera ready version.
>
> **Avg. error values on `case118` with different numbers of supervised samples for Sandwich BNN SvP**
> | N |Train time | %Gap | Max Eq  | Max Ineq  |
> |-|-|-|-|-|
> |512 |15 min|1.5305|0.1041|0.01285|
> |1024|15 min|1.5097|0.0894|0.01442|
> |2048|15 min|1.4691|0.0800|0.01170|
>
> For instance, the Sandwich BNN SvP model achieves a 1.50\% Gap, 0.094 Max Eq., and 0.014 Max Ineq. with 512 supervised samples and a *10-min* training time (avg. of 5 trials). While, with 2048 supervised samples & *15-min* of training, it improves to a 1.46\% Gap, 0.080 Max Eq., and 0.011 Max Ineq. Detailed results can be found at: [Link](https://shorturl.at/nFTE9)
>
> **2. Non-Uniform Performance** We acknowledge the non-uniform performance in certain experiments. We find that this is due to variations in convergence rates that arise under the tight training time constraint (10 mins) that we have imposed on the system.  For the 118 bus system trained on 1024 and 2048 samples, where increasing the training time from 10 to 15 minutes, the performance improves considerably. See the table above and [Link](https://shorturl.at/nFTE9)
>
> **3. Notation & Definitions**
> We will revise the notation. Here $ \mathcal{D}_{f} $ is constructed from the the unlabeled dataset $\mathcal{D}^u$. As a valid solution must fully satisfy all constraints, it has a true feasibility gap $\mathcal{F}$ of zero (eq. 2 in paper). Thus, we transform $\mathcal{D}^u$ into a labeled feasibility dataset $\mathcal{D}^f$ where  each input $\mathbf{x}_j$ has a corresponding feasibility label of zero, i.e.,  $ \mathcal{D}^f =  ( (\mathbf{x}_j, \mathcal{F}(\cdot, \mathbf{x}_j) = 0))$. As generating input samples is computationally cheap (just random variables within bounds), creating this feasibility dataset $\mathcal{D}^f$ is cheap.  Note that during training, weights $w$ & corresponding output $f_w(x_j)$ are tuned to bring the computed feasibility gap $F(f_w(x_j),x_j)$ close to its true value of $0$.
>
> Also, $ p^m_W $ is the posterior distribution of the BNN weights after $ m$ training cycles, which include both supervised & unsupervised learning steps. This serves as the final posterior distribution used for making predictions.
>
> **Questions**
>
> 1&2. Check response under Weakness 3.
>
> 3. Our experiments suggest that early stopping is essential while alternating between supervised & unsupervised stages. In the limited data setting, waiting for each training phase to converge before switching to the next phase can be detrimental. For instance, spending a lot of time in the unsupervised layer can improve feasibility metrics but can adversely affect the %Gap (cost).
>
> 4. Yes. $f_W(\cdot)$ represents the function that NN computes when the weights are set to $W$. And $f_W(x^t)$ is the output of this function for input $x^t$.
>
> 5. Our method has an imposed time constraint and also early stopping to prevent overfitting. Under these conditions, we do not find that increasing unsupervised samples to have a significant effect on the training out comes. Learning with unsupervised data is expensive, hence significant increasing unsupervised data under short training time constraints can only enhance the performance so much. This aligns with other self-supervised DNN methods that require significant compute (Park & Hentenryck (AAAI 2023)).  We performed a preliminary scaling study on case118 by increasing the unsupervised samples to $2^{12}$ and $2^{13}$ while keeping the number of supervised samples at $512$. We only see marginal improvements with $T_{max} = 600s$.
>
>  ### **Sandwich BNN SvP**
> | M (UnSup)| %Gap | Max Eq  | Max Ineq  |
> |-|-|-|-|
> | $2^{12}$ | 1.5471 | 0.077811  | 0.009252  |
> | $2^{13}$ | 1.54431 | 0.07689 | 0.008866  |
>
> We believe that these results can be improved by changing the training constraints to better exploit more unsupervised data. We leave that study for future work.
>
> 6. See reply to weakness 1.
>
> 7.  We use Gaussian prior having 0 mean & 0.01 variance. We tried 1.0, 0.1 and 0.01 as variance of prior and found 0.01 to be most suitable, although quality of solution was not drastically different (all results show same order of magnitude ). We choose very low variance parameter for likelihood ($10^{-6}$) because we know that the ground truth solution of feasibility layer is a delta distribution zero (as discussed for $\mathcal{D}^f$). So, no tuning was performed on likelihood parameters. Hyperparameter details are provided with supplementary material in config.json format and in code.
>
> 8. See response to point 7.

---

### Decision · Program_Chairs · 2025-05-01

**Decision:**

Accept (poster)

**Comment:**

This paper proposes a semi-supervised BNN approach for building optimization proxies in constrained settings with limited labeled data and short training times. The discussion focused on whether the method generalizes beyond the presented power flow application, the limited amount of labeled data in experiments, and the potential benefits of additional baselines. Nonetheless, reviewers noted that the proposed framework shows convincing improvements in non-convex constrained optimization tasks and provides meaningful uncertainty estimates under tight training budgets.

The authors should modify the paper as highlighted in the discussion for the final version: especially in providing further details on the training setup of sensitivity and include additional examples or baselines wherever possible to strengthen the evidence for their approach versatility.